# Homophilic wiring principles underpin neuronal network topology in vitro

Danyal Akarca[1,2,3]*[†‡], Alexander WE Dunn[4,5†‡], Philipp J Hornauer[6], Silvia Ronchi[6], Michele Fiscella[6], Congwei Wang[7], Marco Terrigno[7], Ravi Jagasia[7], Petra E Vértes[8], Susanna B Mierau[4,9,10], Ole Paulsen[4], Stephen J Eglen[5], Andreas Hierlemann[6], Duncan E Astle[1,8†§], Manuel Schröter[6*†§]

[1]MRC Cognition and Brain Sciences Unit, University of Cambridge, Cambridge, United Kingdom; [2]Department of Electrical and Electronic Engineering, Imperial College London, London, United Kingdom; [3]I-X, Imperial College London, London, United Kingdom; [4]Department of Physiology Development and Neuroscience, University of Cambridge, Cambridge, United Kingdom; [5]Department of Applied Mathematics and Theoretical Physics, University of Cambridge, Cambridge, United Kingdom; [6]Department of Biosystems Science and Engineering in Basel, ETH Zurich, Zurich, Switzerland; [7]NRD, Roche Innovation Center Basel, F. Hoffmann-La Roche Ltd, Basel, Switzerland; [8]Department of Psychiatry, University of Cambridge, Cambridge, United Kingdom; [9]Division of Cognitive and Behavioral Neurology, Brigham & Women's Hospital, Boston, United States; [10]Harvard Medical School, Boston, United States

*For correspondence:
danyal.akarca@mrc-cbu.cam.ac.uk (DA);
manuel.schroeter@bsse.ethz.ch (MS)

†These authors contributed equally to this work

‡Co-lead first authors

§Co-lead senior authors

**Abstract** Economic efficiency has been a popular explanation for how networks self-organize within the developing nervous system. However, the precise nature of the economic negotiations governing this putative organizational principle remains unclear. Here, we address this question further by combining large-scale electrophysiological recordings to characterize the functional connectivity of developing neuronal networks in vitro, with a generative modeling approach capable of simulating network formation. We find that the best fitting model uses a homophilic generative wiring principle in which neurons form connections to other neurons which are spatially proximal and have similar connectivity patterns to themselves. Homophilic generative models outperform more canonical models in which neurons wire depending upon their spatial proximity either alone or in combination with the extent of their local connectivity. This homophily-based mechanism for neuronal network emergence accounts for a wide range of observations that are described, but not sufficiently explained, by traditional analyses of network topology. Using rodent and human neuronal cultures, we show that homophilic generative mechanisms can accurately recapitulate the topology of emerging cellular functional connectivity, representing an important wiring principle and determining factor of neuronal network formation in vitro.

## Editor's evaluation

This valuable study examines the principles according to which neurons connect to each other in vitro. The authors show solid evidence that data could be best explained by the homophillic wiring principle where neurons preferentially connect to neurons within overlapping groups.

## Introduction

During mammalian brain development, neuronal networks demonstrate remarkable self-organization that gives rise to complex topological properties, including a greater-than-random clustering and modular structure (*Sporns and Betzel, 2016*; *Newman, 2006*), the selective occurrence of specific network motifs (*Perin et al., 2011*; *Song et al., 2005*), hierarchies (*Hilgetag and Goulas, 2020*), heavy-tailed connectivity distributions (*Alstott et al., 2014*; *Hagmann et al., 2007*), and richly inter-connected hubs (*Achard et al., 2006*; *Bullmore and Sporns, 2009*). Studies have indicated that these distinctive characteristics likely endow neuronal networks with robustness and the capability to support dynamic functional computations (*Laughlin and Sejnowski, 2003*; *Müller et al., 2020*), however, our understanding regarding the underlying mechanisms and wiring rules that give rise to these features is still incomplete.

Neuronal network development can be characterized across spatial scales (*Betzel and Bassett, 2017*; *Schröter et al., 2017*). At the cellular level, neurons form computational units within circuits. Here, the role of individual neurons can be determined by a combination of factors, such as their laminar location, connectivity, neurochemical sensitivities, and morphology (*Kasai et al., 2021*). During embryonic development, a series of spatiotemporally defined genetic and activity-dependent programs (*Pumo et al., 2022*) regulate the expression of cell-type-specific recognition molecules to initiate axonal and dendritic outgrowth, which ultimately leads to the formation of synapses (*Jain et al., 2022*; *Südhof, 2018*; *Yogev and Shen, 2014*). Although there is now a large body of evidence on the mechanisms of specific guidance cues during circuit formation (*Sanes and Zipursky, 2020*), linking this knowledge to explain the emergence of complex topological features remains challenging.

At the whole-brain level in humans (*Vértes and Bullmore, 2015*), connectivity between brain regions can be inferred via diffusion tensor imaging (DTI) (*Jones et al., 2013*; *Jones and Leemans, 2011*), as myelinated axonal connections, or functional magnetic resonance imaging (fMRI) (*Smith et al., 2013*), as correlated patterns of activity. Inferring connectomes from fetal brains in utero (*Turk et al., 2019*), or from preterm infants (*van den Heuvel et al., 2015*), have confirmed the early presence of organizational hallmarks, such as hubs, a rich-club architecture, and a modular small-world organization. Building on such architectures, studies demonstrate that the functional role and organization of brain regions later on is shaped by their inter-regional connectivity and that a region's inputs during development influences its functional specialization (*Johnson, 2011*). This principle allows brain regions to undergo a spatially organized functional shift, for example, from distinct sensory and motor systems to more integrated connections with association cortices, likely supporting an acceleration in cognitive development (*Dong et al., 2021*).

There is growing evidence that some key organizational properties of nervous systems are conserved across scales, and, in some cases, across species (*Betzel and Bassett, 2017*; *Schröter et al., 2017*; *Allard and Serrano, 2020*; *Betzel, 2020*; *Bogdan et al., 2022*; *Engel et al., 2021*; *Zheng et al., 2020*). Nervous systems both at the macro- and micro-scale, for example, have been shown to entail a canonical pattern of small-worldness (*Bassett and Bullmore, 2017*; *Muldoon et al., 2016*), a rich-club topology (*Harriger et al., 2012*; *van den Heuvel and Sporns, 2011*; *Towlson et al., 2013*) and a modular structure (*Sporns and Betzel, 2016*; *Meunier et al., 2010*). These complex organizational hallmarks allow for functional hierarchies, in which distinct segregated modules perform specialized local computations. While the later may reflect basic representational features of incoming signals, intermediary nodes integrate those signals to code for a more complex representation of the incoming signals (*Lord et al., 2017*).

One prominent explanation for these consistent organizational hallmarks is that they reflect the economics of forming and maintaining connections (*Bullmore and Sporns, 2012*; *Bullmore et al., 2016*; *Zhou et al., 2022*). Given finite available resources, trade-offs between incurred costs (e.g., material, metabolic) and functional value must be made continually by all distributed units to ensure optimal network function. In this view, ideas, such as Peters' rule (*Braitenberg and Schüz, 1998*), which suggest that synaptic contacts simply occur if neurons are close enough in space and if their axons and dendrites overlap, are not sufficient to explain the existence of a specific connection (*Briggman et al., 2011*; *Holler et al., 2021*; *Motta et al., 2019*; *Takemura et al., 2015*). Although spatial embedding and neuron morphology clearly have an impact on cortical network architecture (*Ercsey-Ravasz et al., 2013*; *Udvary et al., 2022*), costly features, such as long-range connectivity hub cells or regions, likely exist because they confer some additive functional value that outweighs the cost of

its formation and maintenance (*Bullmore and Sporns, 2012*). Such principles likely apply not only to the connectivity between brain regions, but also to the cellular and subcellular level (*Laughlin and Sejnowski, 2003*; *Vértes and Bullmore, 2015*; *Udvary et al., 2022*; *Chen et al., 2006*).

If an economic trade-off represents an important principle that guides network development, then it is important to consider the specific mechanisms that determine the outcome of this trade-off. Advances in generative network models (GNMs) provide a formal way of testing competing mechanistic accounts of network formation (*Akarca et al., 2021*; *Arnatkeviciute et al., 2021*; *Betzel et al., 2016*; *Chen et al., 2017*; *Goulas et al., 2019*; *Kaiser and Hilgetag, 2004*; *Klimm et al., 2014*; *Morgan et al., 2018*; *Oldham et al., 2022*; *Vértes et al., 2012*; *Yook et al., 2002*; *Zhang et al., 2021*). These computational models simulate the probabilistic formation of networks over time under specific mathematical rules. For example, recent whole-brain DTI work has shown that structural inter-regional connectivity can be simulated with a GNM (*Akarca et al., 2021*) which uses a simple economic wiring equation balancing connection cost with topological value (*Akarca et al., 2021*; *Betzel et al., 2016*; *Zhang et al., 2021*). These two components together define the probability of connections forming iteratively over time. However, the extent to which such models reflect underlying biological processes remains unclear due to the indirect nature of in vivo imaging.

In the present study, we test whether key economic trade-off rules are also conserved at the cellular scale. Analyses are carried out on spike-sorted, high-density microelectrode array (HD-MEA) recordings that allow us to directly record from individual neurons and track both their activity and connectivity across development (*Müller et al., 2015*; *Schröter et al., 2025*; *Ronchi et al., 2021*). Previous studies have quantified the functional couplings among neurons during spontaneous electrical activity and suggested that the local topological statistics are related to firing properties that may drive neuronal self-organization (*Antonello et al., 2022*; *Blankenship and Feller, 2010*; *Warm et al., 2022*). Here, we expand on these analyses and use functional connectivity inferred for individual neurons tracked over several weeks to probe how spiking patterns of neurons facilitate the implementation of economic wiring. Moreover, we translate prior GNM research at the level of inter-regional brain connectivity (*Akarca et al., 2021*; *Arnatkeviciute et al., 2021*; *Kaiser and Hilgetag, 2004*; *Morgan et al., 2018*; *Oldham et al., 2022*; *Vértes et al., 2012*; *Zhang et al., 2021*; *Betzel et al., 2016*; *Liu et al., 2020*) to the cellular scale, to test whether common generative wiring principles are recapitulated in vitro.

We acquire and analyze HD-MEA network recordings from populations of developing primary cultures (PCs) derived from dissociated embryonic rat cortices, three different lines of cell-type enriched human induced pluripotent stem cell (iPSC)-derived neurons, and sliced embryonic stem cell (ESC)-derived human cerebral organoids (hCOs). We compare the performance of different GNMs, and probe whether they can account for the emerging network topology. We also examine the effect of neuronal plating density (cell/mm$^2$) on network topology and test whether GNMs can recapitulate the local organizational properties of the observed networks. Moreover, by chronically blocking GABA$_A$ receptors, we probe how GABAergic signaling impacts neuronal variability and the subsequent formation of connections in the network. Across four model types, comprising 13 wiring rules, we find that homophilic wiring reliably recapitulates the topology and developmental trajectory of neuronal networks at the cellular scale. Homophily may, therefore, represent an important wiring principle in which local network structure is refined via activity-dependent mechanisms.

## Results

### Tracking developing neuronal networks at single-cell resolution

The datasets of this study comprise recordings of rodent and human neuronal networks that were plated and maintained on high-density multielectrode arrays (HD-MEAs; *Schröter et al., 2025*) as previously described (*Figure 1a*; *Ronchi et al., 2021*; *Bakkum et al., 2013a*; *Schröter et al., 2022*). Our primary analysis focuses on primary rat embryonic day 18/19 cortical cultures (PCs), which were plated at two different densities (sparse plating density: 50,000 cells per well, corresponding to 1000 cells/mm$^2$; n=6 cultures; dense plating density: 100,000 cells per well, corresponding to 2000 cells/mm$^2$; n=12 cultures) and used to follow neuronal network development across several weeks in vitro. *Figure 1* provides an overview of the experimental pipeline. We also analyzed human-induced pluripotent stem cell (iPSC)-derived neuron/astrocyte co-cultures,

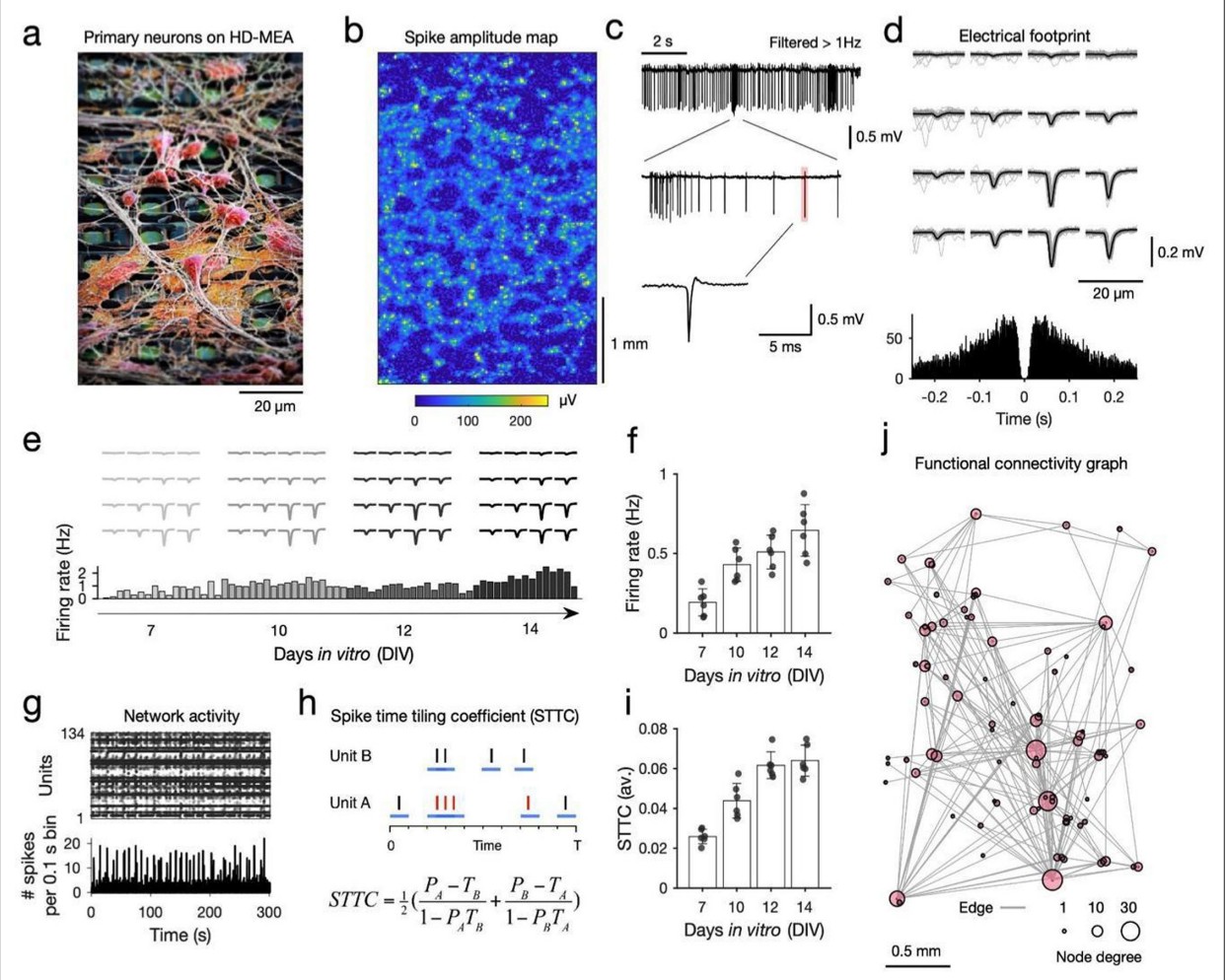

**Figure 1.** High-density microelectrode array extracellular recordings of developing neuronal cultures. (**a**) Scanning electron microscope (SEM) image of primary neurons plated on a high-density microelectrode array (HD-MEA; neurons are colored in red, electrodes in green; the electrode center-to-center distance is 17.5 μm). (**b**) Activity scan across the entire HD-MEA for an example sparse primary cortical (PC) neuron culture at DIV 14. Colors depict the average absolute amplitude of online-detected multi-unit activity per electrode (yellow colors indicate the location of potential neurons, i.e., high amplitude deflection, on the array; the array dimensions are 120×220 electrodes; electrode amplitude values are averaged over 1 min recordings). (**c**) Example extracellular trace recorded from one electrode (10 s), with high action potential (AP) spiking activity and bursts (middle panel); a single AP is depicted in the lower panel. (**d**) Electrical footprint (EF) of a single spike-sorted unit on the HD-MEA. The EF is the AP-triggered extracellular potential signature of a single unit on the HD-MEA, here depicted across 16 electrodes of a high-density recording block in light gray: 100 AP-triggered cutouts for this EF; in black: the average EF. The lower panel shows a spike autocorrelogram for this unit. (**e**) Tracking of individual EFs across development in vitro. The upper panel depicts the EFs inferred for the four recording time points (DIV 7, 10, 12, 14); the lower panel shows the average activity of the tracked unit (bin size: 100 s; gray to black colors correspond to the four recording time points; 30 min per recording day). (**f**) Spontaneous electrical activity of neuronal cultures, their average firing rate, increased significantly with development (n=6 sparse rodent PC networks). (**g**) A spike raster plot shows the spike-sorted activity for one culture (300 s-long zoom in on a network recording with 134 units); the lower panel depicts the binned activity of the same recording (bin size: 0.1 s). All neuronal networks showed a mixture of regular and more synchronized activity periods (network bursts). (**h**) In order to probe neuronal network development, we used the spike time tiling coefficient (STTC) to infer functional connectivity statistically. The four parameters required to calculate STTC connectivity graphs between Unit A and Unit B are $P_A$, $P_B$, $T_A$, and $T_B$. $T_A$ is calculated as the fraction of the total recording time 'tiled' by $\pm\Delta t$ of any spike from Unit A. $P_A$ is the proportion of spikes from Unit A, which lies within $\pm\Delta t$ of any spike(s) of Unit B (spikes in red). The spike trains for Unit A and B are depicted as black bars; $\pm\Delta t$ is depicted as blue lines for each spike. The significance of each STTC value was estimated against jittered surrogate spike train data. (**i**) The average STTC increased significantly with development (n=6 sparse rodent PC networks). (**j**) Example functional connectivity graph inferred from a DIV 14 PC network. Only the top 2% strongest connections are displayed; each dot represents the physical location of a putative neuron on the HD-MEA; the dot size corresponds to the nodal degree of the neuronal unit.

The online version of this article includes the following figure supplement(s) for figure 1:

**Figure supplement 1.** Tracking neuronal networks at cellular resolution on high-density microelectrode arrays.

**Figure supplement 2.** Global topological measures of sparse rodent cultures over time.

containing predominantly glutamatergic, dopaminergic, and motor neurons (plated at a density of 100,000 cells per array), as well as sliced human cerebral organoids (hCOs; n=6 slices). For the complete details of the datasets analyzed in the study, see Methods; *Rodent primary cortical neuronal cultures; Human induced pluripotent stem cell-derived neuronal cultures; Human cerebral organoid slice cultures*. All summary statistics across datasets are provided in *Supplementary file 1a*.

To record the emerging spontaneous neuronal activity of neuronal cultures and to track developing rodent PC neuronal networks at single-cell resolution, we first acquired whole-array activity scans to localize the neurons (*Figure 1b*), and then selected up to 1024 readout electrodes, configured into 4×4 electrode high-density blocks (*Figure 1d and e*), at the respective recording start points (i.e. days in vitro (DIV) 7 for the sparse PCs and DIV 14 for the dense PCs networks). Recordings with the same electrode configuration were acquired at the consecutive recording time points and concatenated for spike sorting (*Pachitariu et al., 2016*). Spike-sorted network data enabled us to assign the extracellular electrical activity to individual neurons and to follow them across development (*Figure 1e*; see also Methods; *Spike-sorting and post-processing*; number of neurons tracked for sparse rodent PC networks: 136±30 (mean±S.D.); number of neurons tracked for dense PC networks: 140±31; *Figure 1—figure supplement 1*). In line with previous works, we find that sparse rodent PC networks developed robust network burst activity (*Figure 1c and g*) and that the firing rate of tracked units increased significantly in the first weeks of development (repeated measures analysis of variance (rmANOVA): F(3,12)=7.02, p=5.62 × 10$^{-3}$; *Figure 1f*).

To infer functional connectivity between neurons statistically and to characterize neuronal network development of tracked neurons over time, we computed the Spike Time Tiling Coefficient (STTC) among all neurons above a minimum firing rate threshold (0.01 Hz; *Figure 1h–j*; *Cutts and Eglen, 2014*) and compared inferred empirical values to jittered surrogate values (see Methods; Functional connectivity inference). We use the STTC because it controls for the variability in firing rates between neuronal units and culture types. STTC values primarily reflect short-latency co-activity rather than firing rates per se (*Cutts and Eglen, 2014*). Although considered a robust measure of pairwise correlations between spike trains, it is important to note that the connectivity graphs inferred by the STTC do not necessarily match the underlying synaptic connections (*Das and Fiete, 2020*). We also apply a recently published transfer entropy (TE) algorithm to show that our results translate to other measures of connectivity (*Shorten et al., 2021*). In line with previous research (*Cabrera-Garcia et al., 2021*; *Walker et al., 2021*), we find that overall STTC values increased with development (F(3,12)=11.82, p=6.77 × 10$^{-4}$; *Figure 1i*), as did the network density of inferred functional connectivity graphs (F(3,12)=11.08, p=8.97 × 10$^{-4}$, sparse rodent PC networks; *Figure 1—figure supplement 2a*). The probability of inferred STTC connections decayed with inter-neuronal distance (*Figure 1—figure supplement 2b*).

## Generative network models of functional neuronal networks in vitro

Following STTC connectivity inference, we set out to describe the topology of these networks using graph theory, which provides a mathematical framework for capturing the topological properties of each node within the network, and the network as a whole. In *Figure 2a*, we highlight three common topological measures (nodal degree, clustering, and betweenness centrality) and one geometric measure (edge length) that we will use throughout the current paper (see Methods; *Network statistics* for more details; *Bullmore and Sporns, 2009*; *Rubinov and Sporns, 2010*). In *Figure 2b*, we show how these statistics allowed us to compute the node-wise statistics for each functional connectivity graph, and to establish the distribution of different statistics across the network.

Although graph theoretic measures provide a way to mathematically formalize the topology of networks, they do not provide an explanation as to what topological attachment principles may have shaped network development. To do this, we tested 13 wiring rules that may best explain the self-organization of cellular-level functional connectivity graphs over time. Each of these 13 rules that we tested is given in *Supplementary file 1b*. To simplify, for the main report, we group the 13 rules into four broader attachment mechanisms by which they work: (i) *Homophily*, where a neuron, *i*, preferentially wires with another neuron, *j*, as a function of the similarity between *i* and *j* in terms of the other neurons they connect to (we expand on this later). (ii) *Clustering*, where neuron *i* preferentially wires with neuron *j* as a function of neuron *i* and *j*'s independently computed clustered connectivity, or (iii)

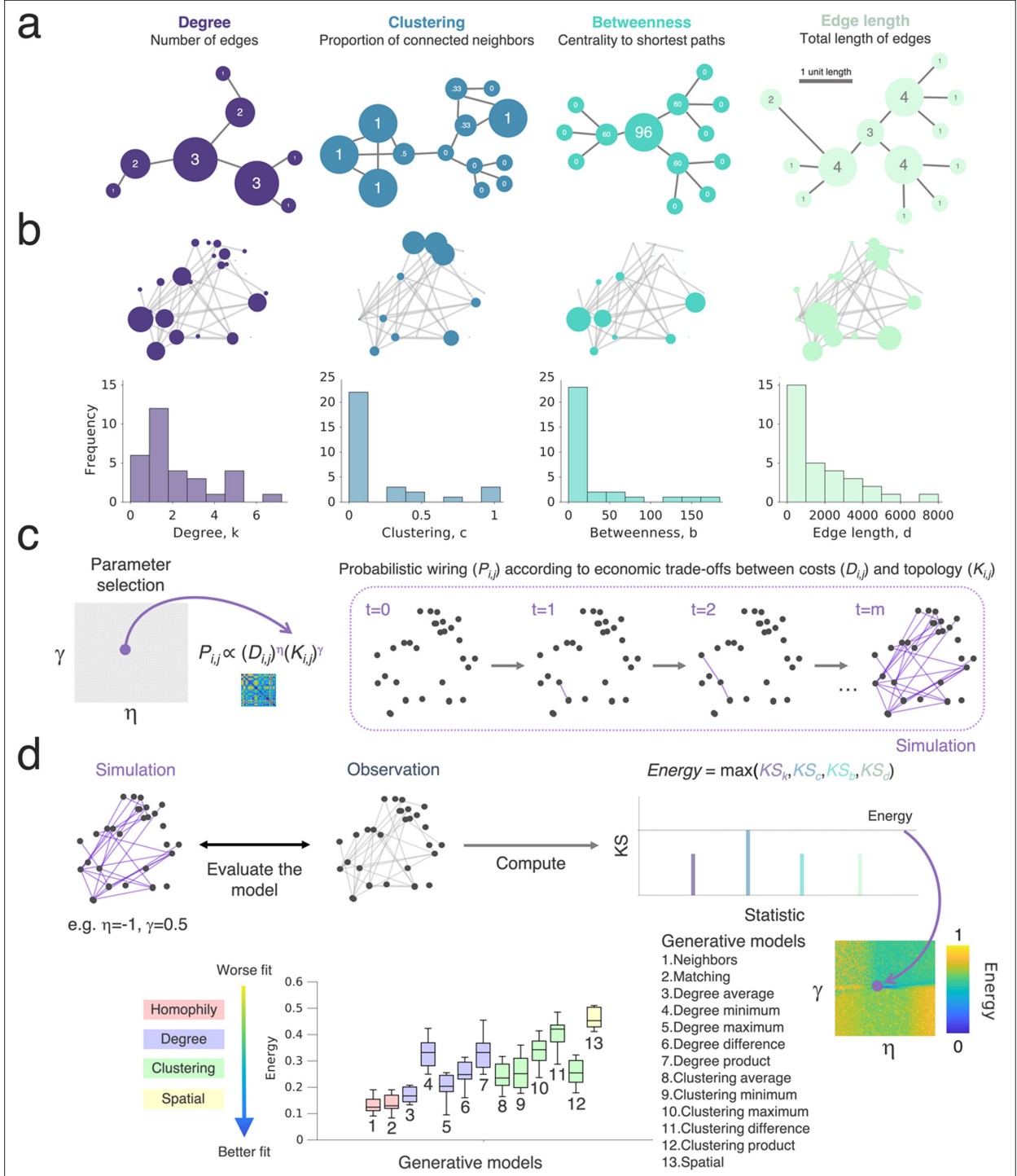

**Figure 2.** Probing wiring principles via generative network models. (**a**) Example networks highlighting four common graph theoretical metrics. For each schematic, the node size corresponds to the respective graph statistic, and we provide the statistic within the node. The panel on the left shows a schematic for the degree (blue), which relates to the total number of edges that each node has. The left-middle panel shows the clustering coefficient (green), which is a measure of how nodes in a graph tend to cluster together; the right-middle panel shows the betweenness centrality (pink), which relates to the number of shortest paths in the network which pass through the node; the panel on the right shows a schematic for the total edge length (yellow), which relates to the total sum of the edge lengths among all its connections for each node. (**b**) Functional connectivity graphs, as inferred from high-density microelectrode array (HD-MEA) network recordings, were characterized by graph theoretical means. Each panel on the top row shows an example schematic network, with the node size corresponding to the respective graph statistic (degree, clustering, betweenness centrality, total edge length). The lower row shows histograms with the distribution of the metric across the network. (**c**) The generative network model works by simulating

*Figure 2 continued on next page*

*Figure 2 continued*

network development probabilistically according to the costs incurred in the network (here reflected by the $D_{i,j}$ term, which is modeled as the Euclidean distances) and the topological values. Both terms are parameterized, meaning that for each simulation, there is an altered tuning in how costs and values influence the probability with which nodes form connections iteratively. Parameters ($\eta$ and γ) alter direction and magnitude to which the costs and values, respectively, influence wiring probabilities. For any combination of wiring parameters, we simulate a network developing over time until it reaches the number of connections, m, equal to the observed empirical network. (**d**) Comparing the simulated and empirically observed network, we fit the generative network model to achieve a simulation that is the least dissimilar to the observation. This is quantified by the energy equation and is shown by dark blue in the parameter space. The energy equation is given by the maximum Kolmogorov-Smirnov (KS) statistic across degree, clustering, betweenness centrality, and edge length. Each parameter combination is plotted on an energy landscape, which demarcates the model fits across the two-dimensional parameter space. Lower energy values correspond to better fits.

The online version of this article includes the following figure supplement(s) for figure 2:

**Figure supplement 1.** Distribution of inter-neuronal Euclidean distances across rodent and human datasets.

*Degree*, the number of their connections. (iv) *Spatial*, where neurons only wire with other neurons as a function of the physical distance between each other.

We start with a model in which *spatial proximity* is the exclusive determining factor for the emergence of a connection. If neurons connect according to Ramón y Cajal's laws of conservation (*Cajal et al., 1995*), balancing the costs of connections with the functional benefits that they provide, one may predict neurons would simply connect to their geometrically closest neighbors to minimize the cost of maintaining connections (*Schröter et al., 2017*; *Bullmore and Sporns, 2012*; *Cajal et al., 1995*). However, other network models incorporating the principle of preferential attachment, such as the Barabási-Albert model (*Barabasi and Albert, 1999*), suggest that a "rich-get-richer" principle may drive the emergence of topology. This is where the more connections a neuron has, the greater the probability of forming more connections - thus leading to scale-free, power law degree distributions (*Yook et al., 2002*; *Barabasi and Albert, 1999*). Therefore, we also apply *degree-based models*, where connections are more likely the more connections one or both neurons have. Another canonical network model, the Watts–Strogatz model (*Watts and Strogatz, 1998*), illustrates how networks deal with the tradeoff between local and global processing—nodes form clusters at the cost of global integration, though local topology is key for modular structure and regional functional specialization, hence are a key component of small-world networks (*Watts and Strogatz, 1998*). We, therefore, include *clustering-based models*. However, clustering-based rules compute how many neighbors of neuron *i* are connected to each other and how many of neuron *j*'s neighbors are connected to each other, separately—they are not necessarily part of the same cluster with the same neighbors. Thus, the benefit of a connection between *i* and *j* may be less than if they had similar neighbors. This could provide a mechanism for neurons with similar functional purposes to connect. Hence, we also include *homophily-based models* based on similarity in neighborhoods between neuron pairs.

We undertook our simulations using a generative network model, which was previously used to probe whole-brain network organization (*Akarca et al., 2021*; *Vértes et al., 2012*; *Betzel et al., 2016*), to determine to what extent these models and findings generalize to the microscale. Generative network models develop in silico networks according to an economic trade-off, in which new connections are iteratively formed depending on both the modeled costs and values (*Figure 2c*). The generative algorithm is expressed as a simple wiring equation (*Kaiser and Hilgetag, 2004*; *Vértes et al., 2012*), which is updated over time:

$$P_{i,j} \propto \left(D_{i,j}\right)^{\eta} \left(K_{i,j}\right)^{\gamma}, \tag{1}$$

where the $D_{i,j}$ term represents the 'costs' incurred between neurons modeled as the Euclidean distance between tracked neurons *i* and *j* (*Figure 2—figure supplement 1*). The $K_{i,j}$ term represents how single neurons *i* and *j* 'value' each other, given by an arbitrary topological relationship which is postulated a priori (also termed 'wiring rule' given mathematically in *Supplementary file 1b*). $P_{i,j}$ reflects the probability score of forming a fixed binary connection between neurons *i* and *j*. This is proportional to the parametrized multiplication of costs and values. Two wiring parameters, $\eta$ and γ, respectively, parametrize the costs and value terms, which calibrate the relative influence of the two terms over time on the likelihood of forming a connection in the next step. We detail the generative network algorithms used in Methods; *Generative network model*. By iterating through different $K_{i,j}$ terms and wiring parameter combinations, we can formally assess how variations in the generative model give

rise to synthetic networks, which are statistically similar to those experimentally observed (*Figure 2d*). Note that while these models are categorized by the network measures they incorporate (e.g. clustering, homophily), they do not necessarily maximize these properties explicitly but rather use them as the mechanism to form connections when balanced against distance. To assess this similarity, the first test comes in the form of an energy function (*Betzel et al., 2016*), which computes the Kolmogorov-Smirnov (KS) distance between the observed and simulated distributions of individual network statistics. It then takes the maximum of the four KS statistics considered so that, for any simulation, no KS statistic is greater than the energy:

$$Energy = max\left(KS_k, KS_c, KS_b, KS_d\right),$$ (2)

where *KS* is the Kolmogorov-Smirnov statistic between the observed and simulated networks at the particular $\eta$ and γ combination used to construct the simulation, defined by the network degree k, clustering coefficient c, betweenness centrality b and Euclidean edge length d. These four measures are critical statistical properties of realistic networks, have been used in prior GNM research to benchmark model fits (*Chen et al., 2006*; *Akarca et al., 2021*; *Vértes et al., 2012*; *Zhang et al., 2021*; *Liu et al., 2020*), and have featured within well-documented simulated network models (*Watts and Strogatz, 1998*; *Albert and Barabási, 2002*; *Beasley and Rodgers, 2012*).

For each empirical network, we simulated 20,000 networks across a wide parameter space (with $\eta$ and γ limits –10 to 10) across the 13 wiring rules and for each network across all available time points. In the present study, we used this wide parameter space as there is little prior work guiding our choice of parameters; we also did not select a seed network. A Voronoi tessellation procedure (*Liu et al., 2020*) was used as the parameter selection algorithm (see Methods; *Parameter selection* for details). Foreshadowing our findings, we report all optimized model parameter fits and energy values across all analyzed datasets in *Supplementary file 1c*.

## Homophilic wiring principles recapitulate the topology of developing rodent neuronal networks in vitro

Previous studies employing generative models of human macroscopic structural brain organization have shown that generative rules based on homophilic attachment mechanisms can achieve very good model fits (*Akarca et al., 2021*; *Oldham et al., 2022*; *Vértes et al., 2012*; *Zhang et al., 2021*; *Betzel et al., 2016*). Homophily-based wiring prioritizes the wiring of nodes preferentially to those with overlapping connectivity patterns (e.g. via neighborhoods or connectivity profiles). For example, under a matching generative model (*Akarca et al., 2021*; *Betzel et al., 2016*), if two nodes have a large proportion of their connections with the same nodes, they will have a correspondingly high matching score because they have similar connectivity profiles. This matching score is *homophilic,* because the measure is defined in terms of similarity (the Greek *homós*, 'same') and preference (*philia*, 'liking'). To test what generative models can best simulate microscale connectivity, we applied the generative procedure to inferred STTC functional connectivity graphs.

We first investigate the sparse rodent PC networks at DIV 7, 10, 12, and 14. As previously shown (*Figure 1*), PC rodent networks underwent significant developmental changes during this time period, reflected by large changes in their functional networks in terms of increasing number of connections and network density (*Figure 1—figure supplement 2*). We find that, over the developmental time course, generative models utilizing the homophilic attraction principle as their generative mechanism produce networks with better model fits (lower energy values) compared to the degree-, clustering-, and spatially based rules tested (*Figure 3a*). By DIV 14, homophily alone performs best (p<4.11 × 10$^{-5}$ for all pairwise comparisons and Cohen's *d*>1.46 reflecting a very large effect size; *Supplementary file 1d* shows all statistical comparisons). The single best performing homophily model, according to the energy equation, was the 'matching model' (see *Supplementary file 1b* for detail), which generates network topology according to the overlapping connectivity between nodes. *Figure 3—figure supplement 1* and *Figure 3—figure supplement 2* shows comparisons across individualized models and across significance thresholds. This model, in contrast to non-homophily models, uniquely generates low model fits and optimized model parameters independent of the increasing correlated activity and changing topology of the networks over this developmental period (see *Figure 3—figure supplement 3* for more details).

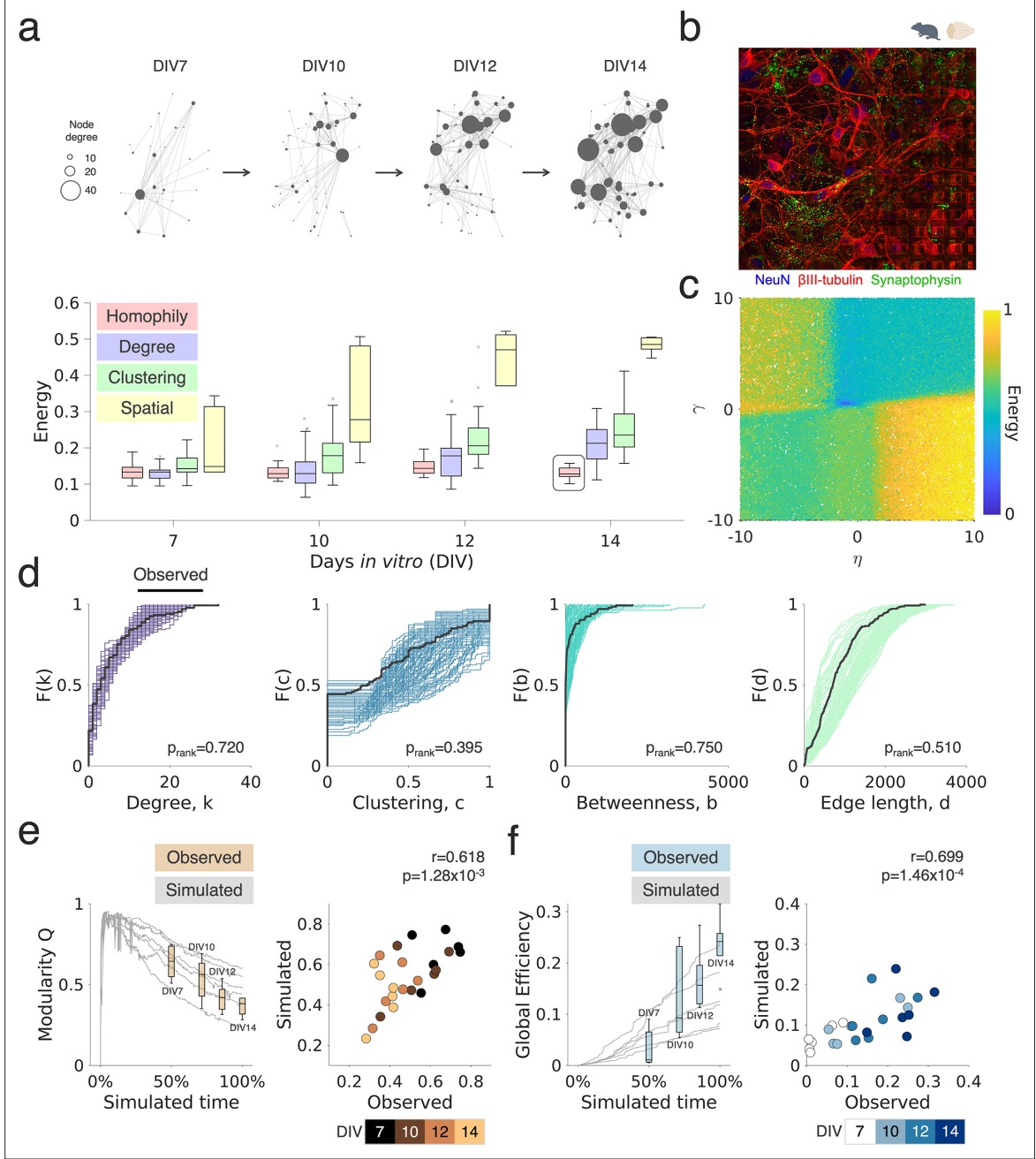

**Figure 3.** Homophilic principles underpin rodent primary network development and explain variance in their maturational trajectories. (**a**) The top row shows a representative sparse rodent primary cortical (PC) network developing over the first two weeks in vitro on the high-density microelectrode array (HD-MEA) (DIVs 7, 10, 12, and 14; sparse neuron plating density; gray nodes correspond to single neurons; the size of each neuron corresponds to its nodal degree). Below, we show the energy of the top performing (n=1) simulated networks, given across the four main categories of studied wiring rules (with all models plotted, grouped by their rule). Boxplots represent the median and interquartile range (IQR); outliers are demarcated as small black crosses and are those which exceed 1.5 x the IQR away from the top or bottom of the box. (**b**) Exemplary immunohistochemical staining of a rodent PC culture (control experiment; NeuN staining in blue, βIII-tubulin in red, and Synaptophysin in green). (**c**) All energy landscapes of DIV 14 networks, plotted on top of each other, under the matching generative network model (a homophily model). (**d**) Cumulative density functions (CDFs) of the matching generative model, showing that simulated and observed statistics overlap very well. CDFs are shown across the four network statistics included in the energy equation of the top = 99 simulations for the matching model (best performing generative model) compared to an observed network (black line). Subsequent $p_{rank}$ values were computed using a Monte Carlo sampling procedure. (**e** and **f**) Simulated network trajectories were examined to determine if the later generative model's trajectory (DIV 14) recapitulated earlier observed statistics. Simulated network statistics were derived by computing the modularity Q statistics (**e**) and global efficiency (**f**), at each time-point, within the generative model that was best fit to the DIV 14 networks. Results were scaled to the age of the culture so that observations and simulations could be compared directly, where 50% of the simulated time corresponds to DIV

*Figure 3 continued on next page*

*Figure 3 continued*

7 and 100% of simulated time corresponds to DIV 14. Note that simulated time here refers to the percentage completion of the simulation, rather than the number of connections added, so that cultures could be aligned based on time. The quoted r and p values correspond to the Pearson's correlation between observed and simulated statistics shown.

The online version of this article includes the following figure supplement(s) for figure 3:

**Figure supplement 1.** Generative network modeling results for the sparse rodent cultures.

**Figure supplement 2.** Generate network model fits as a function of binarization threshold.

**Figure supplement 3.** Generative model outcomes as a function of empirical network data.

**Figure supplement 4.** Alternative generative model procedures.

**Figure supplement 5.** Generative model fits recapitulate observed network statistics.

To assess the extent to which these findings go beyond what may be expected by chance, we further examined model performances across a range of alternative procedures for formalizing the generative models, including a $K_{i,j}$ only formalization (where space has no influence), comparison to density- and size-matched random null models, and a comparison of our networks to those constructed using transfer entropy (TE), which is a model-free effective connectivity measure. We find that, regardless of the model specification or connectivity measure tested here, homophily performs best when approximating the sparse empirical networks relative to all other models but not randomized networks (*Figure 3—figure supplement 4*). For instance, in symmetrized TE networks (i.e. keeping only reciprocal connections), homophily models also perform best (p<1.95 × 10⁻⁶ for all pairwise comparisons and Cohen's d>1.12 reflecting a very large effect size). Furthermore, there is a clear alignment in the generative properties of the networks irrespective of if they were constructed using the STTC or TE, demonstrated by a very large positive correlation in their subsequent model energies (*r*=0.849, p=8.97 × 10⁻²², *r*=0.867, p=9.55 × 10⁻²⁵ and *r*=0.901, p=2.91 × 10⁻²⁹ when considering TE symmetrical, in or out (directed) connections, respectively compared to the STTC functional connectivity graphs; see *Figure 3—figure supplement 4d*).

In *Figure 3b* we show an exemplary immunohistochemical staining of a PC rodent network on an HD-MEA; *Figure 3c* shows the energy landscapes acquired from the matching generative model at DIV 14. The matching generative model, beyond providing the lowest energy values (i.e. very good model fits compared to other models), also produced synthetic networks whose aggregate nodal distributions were statistically indistinguishable from the experimentally observed networks (*Figure 3d*). We formally demonstrated this using a Monte Carlo sampling procedure (*Beasley and Rodgers, 2012*) to directly compare the statistics produced by the well-performing simulations with our empirical observations (degree, $p_{rank}$ = 0.645; clustering, $p_{rank}$ = 0.590; betweenness, $p_{rank}$ = 0.815; edge length, $p_{rank}$ = 0.585; for details of this procedure, see Methods; *Cost functions*). In *Figure 3—figure supplement 5* we provide the same bootstrapping analysis, but for each of the best performing generative models of each model class (spatial, clustering average, and degree average models). It is of note that this matching generative model was the only model that was able to produce statistically indistinguishable results when compared to the experimental observations; the next best performing non-homophily model, the degree average model, failed to approximate both the empirical edge lengths ($p_{rank}$ = 0.025) and participation coefficients ($p_{rank}$ = 0.04).

Next, we asked how well generative models would approximate the time-course trajectories of neuronal network formation. An advantage of the GNM approach is that it allows one to decompose the developmental trajectory. Indeed, if networks are developing according to a homophilic attachment principle then the statistical properties of those simulated trajectories should vary in accordance with our longitudinal observations. To test this, we computed and compared the trajectories of two global network measures of segregation (modularity index Q; *Newman, 2006*) and integration (global efficiency; *Latora and Marchiori, 2001*) across the time course of sparse rodent PC network development. We used these measures because they were both not included in the energy equation (*Equation 2*)—and it is well established that they capture fundamental aspects of how efficient information can be processed across networks (*Damicelli et al., 2019*; *Fair et al., 2007*). Next, we selected the best-fitting model at DIV 14 and decomposed the simulated trajectories up to that point. This allowed us to test whether these simulated trajectories were consistent with the earlier longitudinal observations at DIV 7, 10, and 12.

To do this, we compared each of the longitudinal observations (DIV 7, 10, 12, and 14) to the simulation at the corresponding time point of the DIV 14 developing simulation (i.e. DIV 7, 50%; DIV 10, 71%; DIV 12, 86%, and DIV 14, 100% of the simulated time). *Figure 3e and f* shows the developmental trajectories of modularity and global efficiency for individual simulations (the gray lines) along with the overlaid observed time points. Simulations using the homophily generative model clearly captured the same developmental trend for modularity (decreasing over time) and efficiency (increasing over time) and accounted for a substantial amount of variance in both metrics (modularity: $R^2$=38.2%, $r$=0.618, p=1.28 × 10$^{-3}$; efficiency: $R^2$=48.9%, $r$=0.699, p=1.46 × 10$^{-4}$).

## Topological fingerprints arise from homophilic mechanisms in developing neuronal networks in vitro

We have shown that homophily-based generative models produce synthetic networks which are statistically similar to observed functional rodent PCs networks. However, this similarity depends upon the maximum *KS* distance of the four topological statistics as defined in the energy equation. Crucially, this means that while experimentally observed and simulated network statistical distributions mirror each other at the *global* level, the *topological fingerprint* (*TF*) of these network statistics may still differ. That is, nodes within simulated and observed networks could have different *local* relationships to one another, because node-wise local organizational properties are not captured per se by the existing energy equation. Previous studies have investigated how well generative models can recapitulate local organizational properties and the location of features, such as hub-nodes in the *C. elegans* connectome (*Lindfield and Penny, 2019*) or MRI-inferred human brain networks (*Akarca et al., 2021*; *Arnatkeviciute et al., 2021*; *Oldham et al., 2022*).

This can be exemplified by the topological relationship between central and peripheral nodes. Nodes which score high in centrality measures (e.g. *betweenness centrality*—which determines how many shortest paths pass through—as shown by the red node in *Figure 4a*, left) tend *not* to sit within segregated modules, meaning it is common that they concurrently score low in measures of segregation (e.g. clustering coefficients, in which neighbors connect to each other). The opposite is true for peripheral nodes (see the green node in *Figure 4a*). This means that when correlating measures of centrality with measures of clustering across a network, the correlation tends to be negligible or negative (*Lord et al., 2017*; *Figure 4a*, right).

To assess the ability of generative models to capture these types of local relationships in settings with no anatomical reference space (as neurons are randomly distributed on the HD-MEAs), we provide a very simple cost function, here termed *topological fingerprint dissimilarity* (*TF$_{dissimilarity}$*). The *TF$_{dissimilarity}$* demarcates the ability of in silico network simulations to recapitulate observed local hallmarks of organization. It is defined as:

$$TF_{dissimilarity} = \sqrt{\sum_i \sum_j \left( TF_{observed_{ij}} - TF_{simulated_{ij}} \right)^2}, \tag{3}$$

The *TF* is defined by the n-by-n correlation matrix of n local network statistics for the observed network (*TF$_{observed}$*) and its corresponding (simulated) network (*TF$_{simulated}$*). The *TF$_{dissimilarity}$* is subsequently equivalent to the Euclidean norm (*Lindfield and Penny, 2019*) of the difference between observed and simulated topological correlation matrices. Here, we use six common measures of topology to compute the *TF* matrix (see Methods; *Cost functions* for details). *TF$_{dissimilarity}$* serves as a unitary measure of the difference between the simulated and observed networks in terms of *local* topology, in contrast to the energy metric which reflects *global* topology. *Figure 5a* provides a schematic of how the *TFs* were constructed.

If homophily is a plausible attachment mechanism by which single neurons together form networks, we should expect homophily-based GNMs to produce networks with a local topological structure resembling the observed data. To probe this (dis)similarity, we calculated the *TF$_{dissimilarity}$* between each experimentally inferred functional connectivity graph and the best-performing simulated network (according to the energy equation), for each of the 13 generative models, and across all recording time points (*Figure 4*).

Results demonstrate that synthetic networks generated with homophilic attachment rules provide the lowest *TF$_{dissimilarity}$* from DIV 12 onwards (*Figure 4b*). These rules also result in the statistically

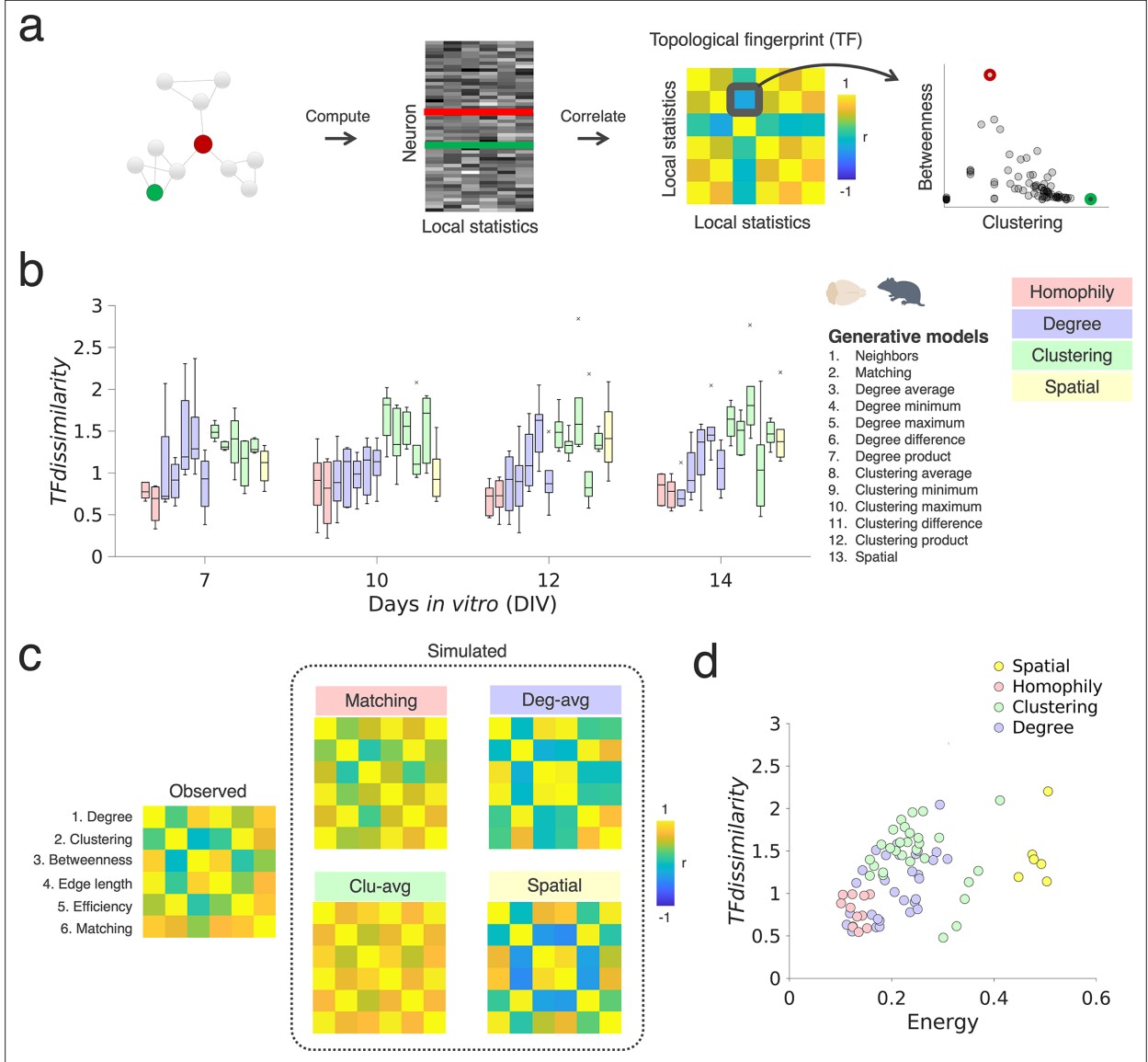

**Figure 4.** Homophilic generative mechanisms capture the local topology in developing neuronal networks. (**a**) Schematic illustrating the inference of topological fingerprints (TFs). A TF measure is computed as the Pearson's correlation matrix among several topological statistics at the nodal level (degree, clustering, betweenness, edge length, efficiency, and matching). Each node in **a** corresponds to a single neuronal unit/neuron. The right panel shows the negative correlation between clustering and betweenness, which captures an aspect of the topological structure of the network shown on the left. The color bar is clipped to +/-1 (blue/yellow) for clarity. (**b**) The $TF_{dissimilarity}$ measures the extent to which GNM simulations capture the topological structure of the experimentally inferred networks. Homophily generative models, and spatial models show the lowest $TF_{dissimilarity}$, suggesting that both can reconstruct local connectivity patterns in vitro (n=6, sparse PC networks; boxplots present the median and IQR; outliers are demarcated as small gray crosses, and are those which exceed 1.5 x the interquartile range, IQR). (**c**) Averaged $TF$ matrix for the empirically observed data (on the left; DIV 14), versus the GNM results obtained from models with the best fits, i.e., the lowest energy values obtained for each model class: the matching rule (top left panel), the clustering-average rule (bottom left panel), the degree-average rule (top right panel) and the spatial mode (bottom right panel). Each node-wise measure used within the correlation matrix is plotted (left). As matching is an edge-wise measure, the matching calculation presented on row six was derived as a node-wise average. (**d**) This plot depicts the relationship between energy and $TF_{dissimilarity}$ values for each sparse PC network broken down by generative rule class (spatial in yellow, homophily in red, clustering in green, degree in blue). Each dot indicates the value of the two model fit functions for a single simulated network.

The online version of this article includes the following figure supplement(s) for figure 4:

**Figure supplement 1.** Homophilic generative mechanisms account for local relationships in developing dense rodent neuronal networks.

**Figure supplement 2.** Relationship between model energy and topological fingerprint dissimilarity in dense rodent neuronal cultures.

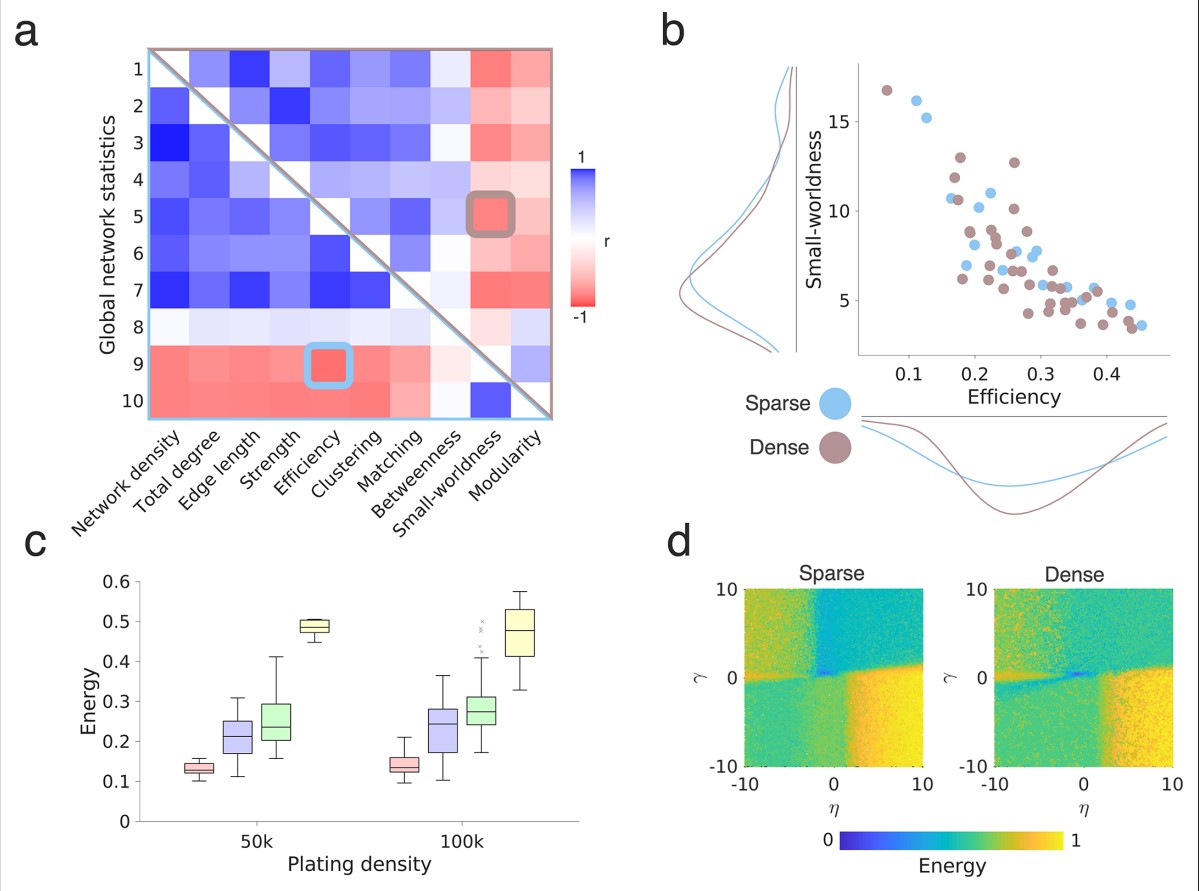

**Figure 5.** Effect of plating density on network topology and generative network principles. (**a**) Correlation matrix for global network statistics calculated across sparse (lower triangle) and dense (upper triangle) primary cortical (PC) cultures shows a highly similar covariance structure; colors indicate the Pearson's correlation coefficient. Elements (5,9) and (9,5) of the correlation matrix are further highlighted in panel. (**b**) This plot shows the correlation between efficiency and small-worldness, for sparse and dense networks ($r=-0.822$, $p=2.73 \times 10^{-14}$); each point in the scatter plot corresponds to a single network. (**c**) Sparse (50,000 cells per well) and dense (100,000 cells per well) PC networks can be best simulated by the homophily generative models (DIV 14). Note that the leftmost boxplot is the same as that given in *Figure 3a*. (**d**) The energy landscapes for the matching generative network model landscapes inferred for the sparse and densely plated PC networks (DIV 14; see also *Figure 3c*).

The online version of this article includes the following figure supplement(s) for figure 5:

**Figure supplement 1.** Comparison of global network statistics across sparse and dense rodent networks at DIV 14.

**Figure supplement 2.** Generative network modeling results for the dense rodent cultures.

smallest $TF_{dissimilarity}$ at DIV 14 (compared to degree rules: $p=1.91 \times 10^{-3}$, Cohen's $d=1.34$; compared to clustering rules: $p=1.37 \times 10^{-7}$, Cohen's $d=2.59$). Of note, homophily and spatial rules could be distinguished significantly at DIV 12 ($p=9.96 \times 10^{-4}$) but not at other time-points (e.g. at DIV 14; $p=0.0952$), possibly indicative of spatial costs driving the topological fingerprint at this level of plating density (*Supplementary file 1e*). Interestingly, replicate analyses in the denser PC rodent dataset provided almost identical results (at DIV 14 and 28), but with an even stronger distinguishability of homophilic rules relative to spatial models (*Supplementary file 1f* and *Figure 4—figure supplement 1*), possibly reflecting a weaker preponderance of costs driving topology. The left-most panel in *Figure 4c* shows the experimentally observed *TF* matrix averaged over n=6 sparse PC networks; the adjacent panels show average *TF* matrices for the matching, clustering-average, degree-average, and spatial generative models. Depicted are the best-performing models within their generative rule class.

In sum, our results highlight the importance of assessing GNM simulation performance both in terms of overall global topology (*energy*) and the local topology generated (*TF*). We find that homophily models concurrently outperform the other models on both fronts (*Figure 4d*, see *Figure 4—figure supplement 2* for a replication analysis in the dense rodent PC network dataset).

## Functional topology and generative model outcome hold in both sparse and dense rodent cultures

So far, we mainly quantified STTC connectivity graphs derived from sparse PC neuronal networks, that is, cultures plated with an initial density of 50,000 cells per well (corresponding to a plating density of ~1000 cells/mm²). Of note, the actual numbers of cells on the HD-MEAs are likely smaller due to incomplete adherence of cells to the HD-MEA and naturally occurring cellular processes such as apoptotic cell death that occur during the first weeks in vitro (*Opitz et al., 2002*; *Xing et al., 2021*; see Methods; *Rodent primary cortical neuronal cultures*). Despite some research on the effect of plating density on the emergence of population activity in vitro (*Wagenaar et al., 2006*; *Ivenshitz and Segal, 2010*; *Cohen et al., 2008)*, synaptic strength and connectivity (*Barral and Reyes, 2016*), there is currently no consensus as to how different plating densities affect neuronal topology. As one critical element of the generative network model is the geometric spacing between neurons, we next probed whether our findings in sparse cultures generalize to networks at higher neuronal plating densities. Therefore, we recorded a second independent dataset of more densely plated rodent PC networks (100,000 neurons per well, corresponding to a plating density of ~2000 cells/mm², n=12; see Methods; *Rodent primary cortical neuronal cultures*) in the exact same way as outlined for the first PC rodent dataset and directly compared both densities at DIV 14.

We found that only the empirical edge lengths and global clustering differed between sparsely versus densely plated PC networks; dense networks showed relatively shorter connections (Mann-Whitney U, p=0.0125) and were more topologically clustered (p=0.0135). All other tested metrics remained very comparable (*Figure 5—figure supplement 1*). The global correlational structure of these statistics also remained stable (*Figure 5a and b*). Given the topological differences in edge lengths and global clustering, we then asked whether this also translated into significant changes in the energy values among the 13 tested generative network models. In *Figure 5c*, we show that the model energy is unaffected by plating density (homophily between plating densities, p=0.191). All statistical comparisons, for each time point in the dense PC networks, are presented in *Supplementary file 1g*; results demonstrate stability not only at DIV 14, but also at a later time point (DIV 28). In *Figure 5—figure supplement 2*, we show the same results, but broken down by each individual model, in addition to showing that this result also holds when considering the average energy over variable best-performing parameter combinations. In *Figure 5d*, we show the energy landscape for both plating densities, which again are very similar.

## Effect of GABA_A receptor antagonism on network development and dynamics

Previous studies have shown that GABAergic interneurons can act as network hubs and regulate synchronicity of spontaneous activity between neurons that is critical for the formation of connections (*Bonifazi et al., 2009*; *Cossart, 2014*; *Hafizi et al., 2021*). The role of GABAergic interneurons during development is also relevant for understanding their function in more mature brain circuits (*Bonifazi et al., 2009*; *Cossart, 2014*; *Le Magueresse and Monyer, 2013*; *Mann et al., 2009*). Moreover, alterations in the ratio of GABAergic interneurons and glutamatergic projection neurons, respectively, the balance of excitation and inhibition, has been implicated in many neurodevelopmental disorders (*Paterno et al., 2020*). Ionotropic GABA_A receptors are known to mediate fast inhibitory transmission in the cortex (in contrast to slow inhibition mediated by metabotropic GABA_B receptors; *Mann et al., 2009*; *Terunuma, 2018*) and are critical for persistent network activity required for behavioral functions (*Paterno et al., 2020*). However, the role played by GABA_A receptors in functional network development is not fully understood. Furthermore, absence of GABA could also delay the developmental switch in GABA polarity from depolarizing to hyperpolarizing (*Ganguly et al., 2001*). Therefore, it is unclear how cellular-scale functional networks would develop in the absence of GABA_A receptor-mediated inhibition and whether connections would form based on homophily as seen in our previous results. It is also unknown whether effects of perturbation to GABA_A receptor-mediated inhibition transmission on network activity and functional connectivity would be reversible.

To address this, we cultured sparse rodent PC neurons under chronic application of media without gabazine (n=6, control) or with gabazine (n=9 gabazine-treated), a selective GABA_A receptor antagonist (see Methods; *Pharmacological experiments*). Following chronic application of gabazine for two weeks, we washed out gabazine at DIV 14 and performed a final recording at DIV 15 in a subset of six

cultures of the gabazine-treated dataset (n=6, termed washout) to determine the extent to which the cultures recovered. We first examine the differences in spiking dynamics and functional connectivity resulting from GABA$_A$ blockade. Next, we examine changes in energy across rules and parameter values within the best-performing rule. We asked whether GABA$_A$ receptor blockade affected network formation by (a) preventing any clear connectivity principle being implemented above the others (where all models have the same energy), (b) altering the connectivity principle being implemented (where a different model than homophily has the lowest energy), or (c) altering how homophily is implemented (where homophilic models still have the lowest energy but parameter directions or magnitudes are altered).

The six control and nine gabazine-treated cultures were compared using Mann-Whitney tests; the six gabazine cultures recorded at DIV 14 were compared to their respective recordings at DIV 15 with Wilcoxon signed-rank tests. In line with previous research (*Xing et al., 2021*; *Baltz et al., 2010*), we find that chronic application of gabazine has a significant impact on single-cell and network firing patterns. Compared to controls, gabazine-treated networks show a more stereotypic burst behavior with less variation in interburst intervals (CV of IBI, Mann-Whitney U: p<0.001; n=6 controls; n=9 gabazine-treated; *Figure 6*). Moreover, we find a trend towards lower firing rates during chronic gabazine treatment at DIV 14 (p=0.0567), which was reversed following washout (DIV 15; Wilcoxon signed rank: p=0.0156; n=6). Burst rates and the fraction of spikes occurring in bursts were highly variable across gabazine-treated networks, and both metrics significantly decreased following washout (p=0.013). We provide a more detailed comparison of the spiking activity between control and gabazine-treated cultures in *Figure 6—figure supplement 1*.

Control and gabazine-treated networks differed in global functional connectivity (*Figure 6—figure supplement 2*). Gabazine-treated cultures showed a reduction in the number of units forming significant functional connections (lower network size, p<0.001, and total degree, p=0.012). Note that following washout, network size can change as only spiking neurons are counted as nodes for network size calculation. Network size did not increase significantly (p=0.0625) following washout, though there was a small increase in network size in five of the six washed out cultures (by 18.4 ± 6.2%). On average, gabazine-treated networks showed an increase in average STTC (Mann-Whitney U; p<0.001), a higher network density (p=0.003), and a greater total edge length (p=0.026). Following the washout, average STTC (p=0.031), network density (p=0.031), and edge length (p=0.013) reduced significantly, resembling the untreated cultures. Whereas untreated cultures showed a small fraction of nodes with high nodal strength, indicative of potential hub nodes, we did not see this topological structure during chronic gabazine treatment. That is, control networks had a more positively skewed STTC distribution than controls (p=0.002). This is consistent with the notion that GABA$_A$ receptors are involved in the regulation of spiking activity and synchrony between neurons in the network, as this was altered by GABA$_A$ receptor block.

To further examine the extent to which washing out gabazine returned endogenous inhibitory activity as GABA$_A$ receptors were released from blockade, we also computed the spike transmission probability (STP) among neuronal units (*Figure 6—figure supplement 3*). This cross-correlogram-based metric has been used to identify spike transmission, respectively spike suppression, and to infer putative excitatory/inhibitory functional connectivity from on-going extracellular spiking activity (*Spivak et al., 2022*). Indeed, we find that washout of gabazine led to a significant increase in the number of putative inhibitory connections (Wilcoxon signed rank: p=0.012; *Figure 6b*). It should be noted, however, that weaker inhibitory connections are likely missed with the applied STP approach (*English et al., 2017*).

Despite alterations in cellular activity and functional network topology (*Figure 6—figure supplement 1* and *Figure 6—figure supplement 2*), homophilic generative attachment rules were the best fitting models across both control, gabazine-treated, and washout conditions (*Figure 6—figure supplement 4a and b*). However, gabazine cultures exhibit a higher energy relative to controls (p=0.0292; *Figure 6c*), which suggests that homophilic GNMs cannot approximate the functional topology of gabazine-treated cultures to the same extent as for control cultures. This finding supports our hypothesis that perturbing GABA$_A$ receptor-mediated inhibition alters functional network characterization, even though homophily remains the best-fitting model. After washout of gabazine, however, these same cultures exhibited homophily energy values indistinguishable from controls (p=0.285), which indicates that this effect may be reversible. *Figure 6—figure supplement 4c* shows how gabazine

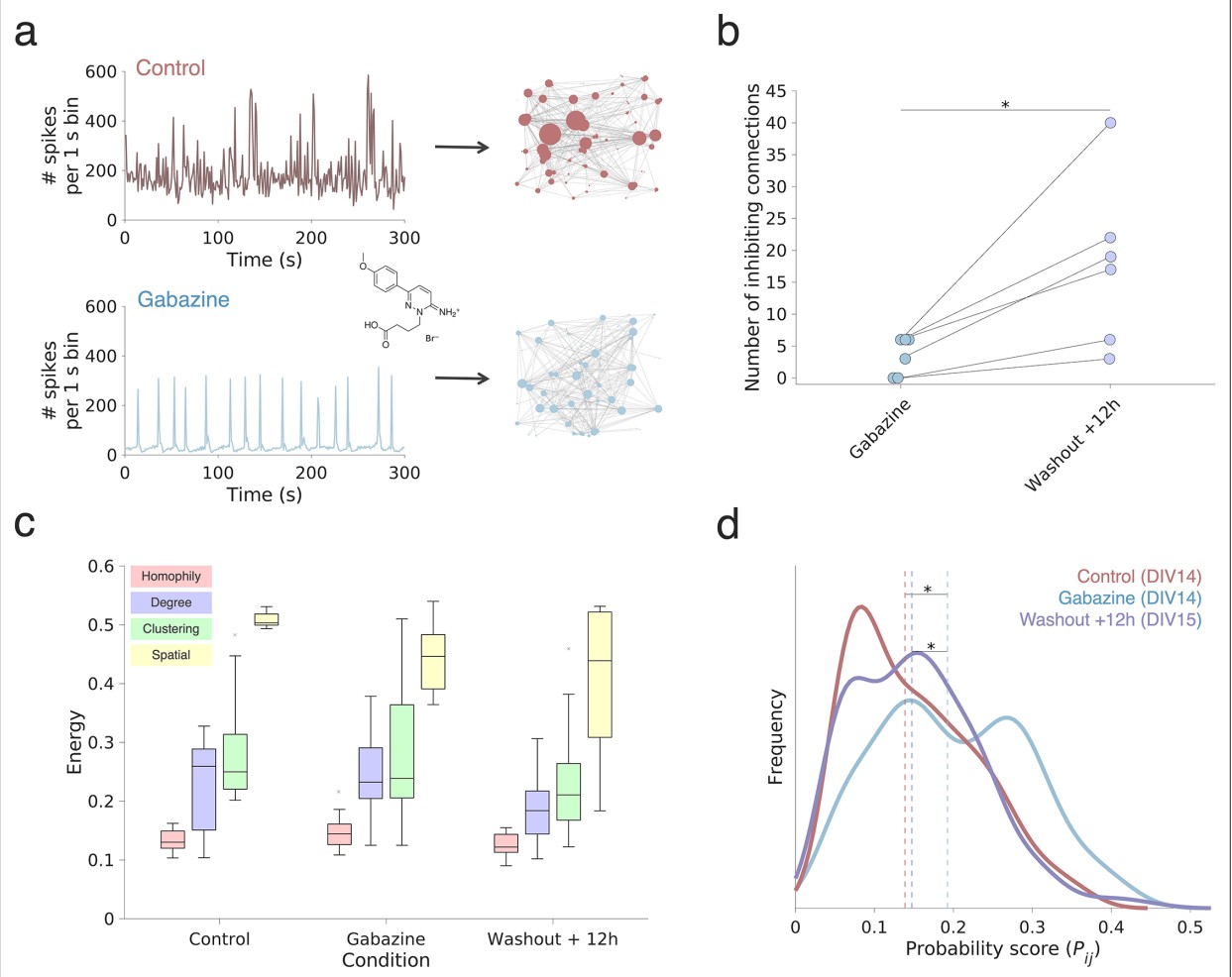

**Figure 6.** Chronic block of GABA$_A$ receptors alters the wiring parameters of the homophily generative model. (**a**) Representative neuronal population activity of a control/untreated (red, top) and gabazine-treated networks (bottom, blue) at DIV 14. In the gabazine condition, network burst activity is more strongly synchronized compared to the control networks. (**b**) Panel depicts that the number of putative inhibitory connections increased significantly following the washout of gabazine at DIV 14. Putative inhibitory connections were inferred by a cross-correlogram approach and using an algorithm to infer spike transmission probability, respectively, spike suppression (***English et al., 2017***). (**c**) The energy of the top-performing (n=1) simulated networks, given across the four main categories of studied wiring rules. Boxplots represent the median and interquartile range (IQR); outliers are demarcated as small black crosses and are those which exceed 1.5 x the IQR away from the top or bottom of the box. Energy values and wiring parameters for each individual rule of the top performing (n=1) simulated networks across control, gabazine-treated, and washout cultures are shown in ***Figure 6—figure supplement 4c***, respectively. (**d**) The average probability score kernel density distributions ($P_{ij}$) for the control (brown), gabazine (cyan), and washout conditions (lilac) recorded in sparse primary culture networks across simulated network development. The average gabazine probability distribution is shifted to the right and flatter compared to the controls and the washout condition. This indicates that wiring becomes more random across all time points. In ***Figure 6—figure supplement 4d*** we show all distributions used to construct these averages. Asterisk indicates p<0.05.

The online version of this article includes the following figure supplement(s) for figure 6:

**Figure supplement 1.** Chronic gabazine application to block GABA$_A$ receptor activity led to changes in spiking patterns.

**Figure supplement 2.** Chronic gabazine application to block GABA$_A$ receptor activity led to changes in functional network topology.

**Figure supplement 3.** Inferring inhibitory connections using spike-transmission probabilities.

**Figure supplement 4.** Comparison of generative model outcomes between control, gabazine-treated and washout conditions.

cultures are best fit when the $\eta$ homophily wiring parameter increases so that it is closer to zero, i.e., moving from more negative to less negative, decreasing in magnitude (p=0.0360). Lower magnitude $\eta$ corresponds to a weaker influence of physical distance on the probability of forming connections. On the other hand, γ (which varies the extent to which homophily influences wiring probability) is

unaffected (p=0.456). Hence, gabazine weakens the spatial extent of functional connectivity wiring, enabling longer-distance connections in the network.

Differences in wiring parameterization may reflect more fundamental differences in neuronal variability elicited via changes in the neural dynamics. Within the homophily generative model, connections form via continual negotiations between the modeled cost and self-similarity that is present between all neurons. If there are clear *relative* winners in this negotiation, for example, connections that are lower cost than all others and connections that are more homophilic than all others, these connections are more likely to form. Of note, the probability of a connection is proportional to the wiring parameter magnitudes (see *Equation 1*). Mathematically, the $P_{i,j}$ distribution would look like a canonical lognormal distribution that is found at many anatomical and physiological levels of the brain (*Buzsáki and Mizuseki, 2014*; *Loewenstein et al., 2011*), where a small number of possible candidate connections elicit a higher wiring probability while the majority remains low. However, in the example of gabazine, neuronal synchrony increases, which is equivalent to exhibiting less variability in spike times between neurons. Therefore, we hypothesized that this would lead to a flattening of this lognormal distribution, meaning that the resultant topology became more random (see Methods; Generative probability distributions for details). Indeed, in *Figure 6d* we show this to be the case: gabazine-treated networks exhibit a flattened $P_{i,j}$ distribution relative to both controls (median $P_{i,j}$ value = 0.135 and 0.322 for gabazine & control, respectively; p=1.54 × 10$^{-44}$, Cohen's $d$=0.550) and also after gabazine washout (median $P_{i,j}$ value = 0.179; p=5.04 × 10$^{-8}$, Cohen's $d$=0.196; see also *Figure 6—figure supplement 4d*). This finding suggests that gabazine alters the network wiring distribution as to become more variable in its wiring preferences, rather than being specific to a relatively smaller number of candidate neurons that are deemed particularly valuable to wire with.

In summary, we find that homophily wiring rules are also the best fitting GNM for neuronal networks chronically treated with gabazine. For the latter, however, homophilic GNMs achieve worse fits compared to control cultures. At the level of model parameters, the homophily γ parameter remains stable over all conditions, but it is $\eta$ - which alters the spatial extent over which wiring is constrained. Interestingly, following washout of gabazine at DIV 14, numerous topological characteristics are recovered. Within our simulations, the washout effect is reflected as a shift towards a more canonical lognormal connectivity distribution (*Buzsáki and Mizuseki, 2014*).

## Probing generative wiring principles across different human neuronal networks

As shown in the previous section, patterns of neuronal spiking dynamics may have an effect on how rodent neurons form functional connectivity in vitro. However, to what extent this idea translates to, for example, networks comprising specific kinds of human neurons and their respective/varying spiking dynamics (*Bakken et al., 2021*; *Berg et al., 2021*) remains unclear. We start to address this question by applying GNMs to purified human iPSC-derived neuron/astrocyte co-cultures. GNM analyses were performed at a time point at which such cultures reach a state of relative maturity (DIV 28) (*Ronchi et al., 2021*). The dataset for this analysis comprises purified glutamatergic neurons (GNs, n=8), motor neurons (MNs, n=7), and dopaminergic neuronal cultures (DNs, n=6). We also included slice cultures derived from 4-month-old human embryonic stem cell-derived cerebral organoids (hCOs, n=6 slices). Previous studies have indicated that hCOs develop functional networks with increasing complexity from as early as 90 days in vitro (*Giandomenico et al., 2019*; *Szebényi et al., 2021*). *Figure 7a* shows an immunohistochemical staining of a DIV 21 human iPSC-derived DN culture, expressing neuronal and astrocytic markers (MAP2, GFAP, and TH); *Figure 7b* shows stainings for hCOs slices (Tau, NeuN, and GFAP).

*Figure 7c* provides an overview of the human and rodent neuronal spiking dynamics. Following a t-distributed stochastic neighbor embedding (tSNE) analysis, we find that networks can be clustered well according to their spiking dynamics. The tSNE analysis is based on spike-train autocorrelograms, derived from the aggregated activity of each neuronal network (*Petersen et al., 2021*; see Methods; *Autocorrelogram analysis*). Example network activity plots, illustrating the different firing dynamics and corresponding autocorrelograms, are shown in *Figure 7d*; *Figure 7—figure supplement 1a*. Overall, activity-based tSNE clustering allows grouping of different monolayer human neuronal cell lines in space; greater spread/variability is observed for the hCO slices. MN dynamics appear closest to rodent primary cortical neurons (PCs) in the tSNE space, which is reflected by their similarity

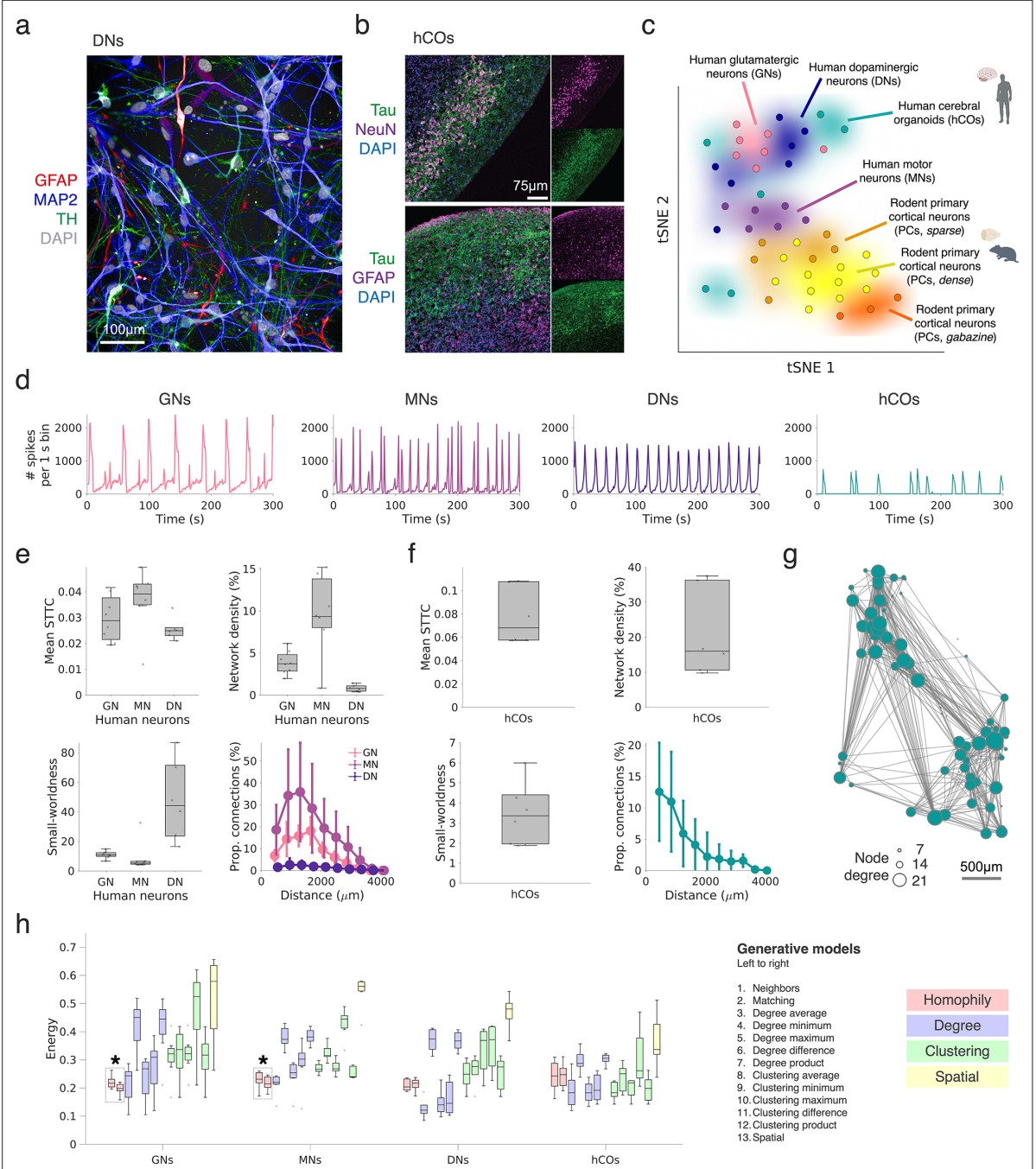

**Figure 7.** Generative network modeling across human monolayer neuron cultures and cerebral organoids. (**a**) Immunohistochemical staining of a control dopaminergic neuron (DN) network (top panel: expression of GFAP in red, MAP2 in blue, TH in green, and DAPI in gray). (**b**) Immunohistochemical staining of a single human cerebral organoid (hCO) slice (Tau in green, NeuN in purple and DAPI in blue; bottom panel: Tau in green, GFAP in purple, and DAPI in blue). (**c**) Clustering of human and rodent neuronal cultures based on a t-distributed stochastic neighborhood embedding (tSNE). Each dot corresponds to one culture; similarity was estimated by correlating the spike train autocorrelograms across all datasets and by then applying the tSNE. (**d**) Representative population activity plots for networks of glutamatergic (left), motor (left middle), dopaminergic (right middle) iPSC-derived neuronal cultures and a hCO slice (right). (**e**) Global topological measures of human iPSC-derived neuronal networks at DIV 28: the mean spike time tiling coefficient (STTC) (top left), network density (top right), small-worldness (bottom left) and proportion of extant connections by distance (bottom right). (**f**) STTC functional connectivity measures and small-worldness of hCO slices including: average STTC (top left), network density (top right), small-worldness (bottom left), and connectivity as a function of distance (bottom right). (**g**) Functional connectivity graph inferred from a hCO slice. (**h**) Fits across generative model rules in human iPSC-derived neuronal networks (DIV 28) and hCOs (right). The energy of the top n=1 performing simulated networks are shown. Boxplots represent the median and interquartile range (IQR). Outliers are demarcated as small black crosses and are those which exceed

*Figure 7 continued on next page*

*Figure 7 continued*

1.5 x in the IQR away from the top or bottom of the box. The asterisk reflects homophilic rules, which demonstrate statistically lower energy (p<0.05) than other rules. All statistics are provided in ***Supplementary file 1h***.

The online version of this article includes the following figure supplement(s) for figure 7:

**Figure supplement 1.** Spiking dynamics in human monolayer neuron and cerebral organoid cultures compared to rodent cultures.

**Figure supplement 2.** Replication of the main results using an alternative spike-sorting and postprocessing pipeline.

**Figure supplement 3.** Replication of the main results on a validation dataset.

in ongoing spiking dynamics (***Figure 7—figure supplement 1b***). Representative examples of the observed differences in population activities across different human neuron cultures are depicted in ***Figure 7d***.

As for the rodent cultures, we constructed STTC functional connectivity graphs for all human neuron cultures and assessed key connectivity metrics and topology. ***Figure 7e*** and ***Figure 7f*** show these network statistics, respectively; ***Figure 7g*** provides an example of a functional network obtained from a hCO recording. It is notable that human iPSC-derived neuronal networks did not differ significantly in average STTC (ANOVA, p=0.0912). However, we find a significant difference in network density (p=$1.02 \times 10^{-4}$) and topological metrics, such as the small-world index (p=$1.51 \times 10^{-4}$). MNs show the highest network density, as inferred by the STTC, and small-worldness values within the range previously observed for the rodent PC networks.

***Figure 7h*** depicts the GNM results across all human monolayer and hCO cultures. Overall, generative model findings approximately mirror the similarity in their underlying spiking dynamics (***Figure 7d***), in that human MNs show an energy profile across generative models aligning with rodent cultures, followed by GNs (***Figure 7h*** left and middle-left), homophilic rules p<$3.70 \times 10^{-3}$ for all pairwise comparisons and Cohen's d>0.772 reflecting large effect sizes. In contrast, DNs and hCOs, which show differing underlying dynamics, exhibit a very distinct profile with the degree average method achieving the lowest energy, but no non-spatial class of generative rules providing the statistically lowest energy (***Figure 7h***, middle-right and right). As noted previously, hCOs exhibited significant variability in their spiking dynamics and topology, and provided an energy profile somewhat resembling randomly rewired graphs (as in ***Figure 3—figure supplement 4***, right). All statistical findings are provided in full in ***Supplementary file 1h***.

In summary, GNMs based on homophilic wiring mechanisms best recapitulate functional network connectivity in vitro in rodent PC networks, particularly at later developmental stages (DIV 14 onwards). We replicate our main findings using alternative spike-sorting and postprocessing pipelines, and validate them on an independent dataset in ***Figure 7—figure supplements 2 and 3***, respectively. In a few cases, including some immature rodent PC networks and human neuron cultures, the degree-average-based model performs well, if not best. This may reflect differences in the underlying spiking dynamics and/or parameter choices during functional network inference (***Figure 7—figure supplement 1***). Human GN and MN networks show the closest resemblance to rodent PC networks and as such, mirror their underlying homophilic generative mechanisms. Results in hCOs are as yet inconclusive, likely due to the observed variability in their spiking activity, the limited number of consistently active units per network, and the prolonged developmental timeline for establishing functional connectivity.

## Discussion

In the current study, we applied large-scale electrophysiological recordings to track and characterize single-unit functional connectivity as neuronal networks develop in vitro. We systematically tested which candidate topological attachment mechanisms could explain this developing self-organization, using a range of generative network models to simulate network formation in silico. In the majority of cases tested, we found that a model utilizing a homophilic attachment mechanism (***Holler et al., 2021***) performed best. This model accurately captures the developmental trajectories of neuronal networks, their local topological organization, and highlights how neuronal variability is likely critical to the emergence of the canonical lognormal distribution (***Buzsáki and Mizuseki, 2014***) of neuronal functional connectivity within networks. The apparent symmetry between these findings and previous work at various scales of analysis (***Akarca et al., 2021***; ***Oldham et al., 2022***; ***Vértes et al., 2012***;

*Zhang et al., 2021*; *Betzel et al., 2016*) and species (*Chen et al., 2006*; *Carozza et al., 2023b*) may have implications for the study of brain development.

## Topological self-similarity as a driver of in vitro topology

In line with previous work, we find that functional connectivity increased with development, and that developing neuronal networks in vitro exhibited typical characteristics of complex network architecture seen across numerous scales (*Schroeter et al., 2015*; *Downes et al., 2012*). Our work demonstrates that, beyond the other tested models, homophily best recapitulates complex functional network topology in vitro.

Interestingly, we find that the degree-average model tends to consistently follow the performance of homophily in cultures that are relatively less complex (e.g. immature rodent cultures and human single-cell-type cultures) but do worse relatively later in development (e.g. *Figure 3—figure supplement 3*). Conceptually, the degree-average generative model prefers connection formation when both neurons simultaneously have large numbers of connections (given that $\gamma > 0$). Importantly, the degree-average model (as with all other models apart from homophily) treats the underlying computation for how networks form connections as entirely independent. That is, the computation for the probability to wire is made not with direct respect to any other node pair—it is made on each neuron before then performing some other operation (e.g. taking the average or maximum). In contrast, homophily is a function made with respect to the *direct* relationship between the connections of the neurons. This is a subtle but important theoretical distinction, because it highlights how homophily can occur via the local communication between neurons, where signals are propagated via their connections. This is likely critical, as any generative mechanism by which complex neurobiological networks develop are likely to emerge from these interactions between the local components over time (*Johnson, 2011*)—without a central mechanism aiming to optimize its global network properties (*Malkov and Ponomarenko, 2016*). It is important to note, however, that the present study focuses on functional connectivity, rather than structural connectivity, and this separation is discussed in more detail later in our limitations.

A related observation is that the homophily heuristic—much like in social networks (*McPherson et al., 2001*; *Talaga and Nowak, 2020*)—enables each part of the network to interact with its local environment without requiring inordinate computational resources. Indeed, homophily has been shown to provide an efficient trade-off capable of producing small-world networks through which information can propagate efficiently (*Malkov and Ponomarenko, 2016*). Under this view, as limits to *local knowledge* and *computational capacity* hold for any interacting developing system, homophily becomes a generative heuristic for any sufficiently large network. Notably, this resonates with biological accounts of Hebbian learning (*Hebb, 1949*; *Vértes et al., 2014*) and spike-timing dependent plasticity (STDP) (*Song et al., 2000*) whereby neurons wire with each other as a function of *similarity* to themselves (e.g. concurrent or temporally precedent neuronal firing, respectively) provided that neurons are sufficiently close in space (*Song et al., 2014*; *Liu et al., 2024*). This also resonates with computational accounts of other developing networks, such as the internet or metabolic networks, that use hyperbolic maps to define node attractiveness in terms of both 'popularity' (e.g. nodes having a large number of connections) and 'similarity' (e.g. being physically proximal or having similar connectivity profiles) (*Papadopoulos et al., 2012*). These representations of networks have the benefit that nodes can efficiently route information across the network using only the coordinates of their neighbors (*Boguñá et al., 2010*); emphasizing both the importance of including geometry in network representations and that homophily has a core role in network organization.

## Models of developing networks across scales, species, and time

The combination of generative modeling and graph theory allows us to use in silico simulations as a lingua franca to probe micro-connectomic self-organization (*Schröter et al., 2017*). Comparative studies have examined economic accounts of connectomic organization across different species (*Vértes and Bullmore, 2015*)—such as in the worm *C. elegans* (*Towlson et al., 2013*; *Nicosia et al., 2013*; *Witvliet et al., 2021*), larval zebrafish (*Betzel, 2020*), mouse (*Johnson et al., 2019*), macaque (*Chen et al., 2017*) and human connectome (*Akarca et al., 2021*; *Arnatkeviciute et al., 2021*; *Betzel et al., 2016*; *Kaiser and Hilgetag, 2004*; *Klimm et al., 2014*; *Morgan et al., 2018*; *Oldham et al., 2022*; *Zhang et al., 2021*; *Liu et al., 2020*). For example, *Nicosia et al., 2013* modeled the growth

of *C. elegans* using the known birth times of its somatic neurons—finding that as the body of the animal progressively elongates, the cost of longer-distance connections become increasingly penalized. In human brain scans, *Oldham et al., 2022* incorporated known early changes in brain macroscopic geometry and other physiological measures of homophily (e.g., correlated gene expression) to improve an additive generative model's network embedding (also see *Arnatkeviciute et al., 2021*). These works have highlighted the benefit of incorporating specific developmental changes, that are specific to the organism, within a growth model able to simulate developmental outcomes.

## Homogeneous spiking dynamics may lead to stochastic wiring through reduced parameter magnitude

In line with previous findings, the inhibition of GABA$_A$ receptors led to increased neuronal synchronization compared to untreated cultures (*Mann et al., 2009*; *Assenza et al., 2011*). Despite this difference in population dynamics, the homophily generative model better reproduced network topology than the other models, albeit with less error in the control networks. Crucially, GABA$_A$ receptor block did significantly push the model parameter, $\eta$, closer to zero. Reduced $\eta$ magnitude indicates a weaker preference for short-distance connections in gabazine-treated networks compared to controls, whilst still preferring shorter connections, as indicated by $\eta$ being negative. One explanation might be that increased synchronicity with gabazine application leads to functionally less distinguishable spike dynamics between neurons—hence, the connection probability distribution was flatter and shifted to the right relative to the probability distribution of controls. Given this shift, it could be expected that simply spatial proximity would better explain connection formation. However, $\eta$ magnitude decreased rather than increased with gabazine treatment. This indicates that gabazine-treated cultures showed a weaker rather than stronger preference for short-distance connections compared to controls. One explanation could be that gabazine-treated cultures fail to deselect less homophilic long-distance connections. Consistent with this, gabazine-treated cultures showed a longer total edge length, reflecting more long-distance, high-cost connectivity than controls. Together, these results support that GABA$_A$ receptor-mediated activity influences connection specification that can elicit the same network topology more efficiently by maximizing value relative to the cost of connections.

Interestingly, previous GNM work at the whole-brain scale has shown that lower magnitude wiring parameters are associated with poorer cognitive scores (*Akarca et al., 2021*, age *Akarca et al., 2021*; *Betzel et al., 2016*) and a diagnosis of Schizophrenia (*Vértes et al., 2012*; *Zhang et al., 2021*). Perhaps, inhibitory hub neurons fail to implement the principle of homophily in the same way, thus constraining the emergence of small-worldness which has also been related cognitive function (*van den Heuvel et al., 2009*; *van den Heuvel and Sporns, 2013*). This may suggest convergent evidence for how developmental stochasticity, intrinsic to how developing parts interact with each other, may influence functional outcomes (*Hiesinger and Hassan, 2018*; *Carozza et al., 2023a*). A remaining challenge in the field is to be able to directly parse the extent to which stochasticity and specific economic trade-offs may influence network outcomes under different conditions.

## Limitations and future work

Currently, there is a lack of consensus as to how functional connectivity can be inferred from the spontaneous activity of neurons developing in vitro (*Abril et al., 2018*). In the present study, we utilized the spike time tiling coefficient (STTC; *Cutts and Eglen, 2014*) which was developed to improve on some of the limitations of traditional metrics for the coupling between neurons, such as the correlation index (*Wong et al., 1993*). Crucially, our goal was not to infer synaptic connections, direct structural connections nor reconstruct the underlying circuit of synaptic (and autaptic; *Deleuze et al., 2014*) connections present. Rather, we sought to infer relationships in temporal patterns of spiking activity between neuronal units that may be relevant to network emergence and its functioning. Thus, the present analysis sits at a larger spatial scale than the synaptic or circuit level within the spatial and temporal structure of the brain. Nevertheless, in *Figure 3—figure supplement 4*, we show that our results resemble a TE-based method (*Shorten et al., 2021*; *Novelli et al., 2019*; i.e. measures of effective connectivity), which may better correspond to how information flows between neurons in a network. Furthermore, similar results were found in standard density MEAs, though neuronal units were not tracked over time (*Antonello et al., 2022*). Nevertheless, it is important to keep in mind how our presented results and interpretations may or may not generalize to structural connectivity graphs

of in vitro neuronal networks directly, despite the same rules approximating macroscopic structural connectomes (*Akarca et al., 2021*; *Betzel et al., 2016*; *Goulas et al., 2019*; *Oldham et al., 2022*). For example, the type of cost incurred by the existence of a long-range functional connection is not necessarily equivalent to that incurred by a structural connection, as a direct physical connection may not exist. Future work is needed to investigate directly structural connectivity at this scale and examine the extent to which there are equivalences between the formation and development of structural components (e.g. synapses and axons) alongside their functional interactions.

Another important consideration is the impact of subsampling, and specifically, how the selection of HD-MEA recording configurations and neuronal units (e.g. including units only above a specific activity threshold) impacts the reported organizational properties in functional connectivity. Previous studies established that subsampling can significantly affect the inference of network dynamics and connectivity (*Das and Fiete, 2020*; *Pillow and Latham, 2007*), and this should be considered when interpreting our results. Future studies could further probe how different parameter choices for selecting HD-MEA recording configurations and the exclusion of low activity units alter the inference of neuronal topology at this scale. Similarly, it would be important to test if the reported results also hold for more advanced network inference algorithms (*Donner et al., 2024*).

Moreover, an important outstanding question is also how the role of $GABA_A$ receptor-mediated activity changes with development, and whether there is any link to the observed findings. From DIV 7 to DIV 14, homophily became more distinguishable from other rules in terms of its recapitulation of network topology in rodent PC networks. This coincides with the gradually increasing proportion of GABAergic synapse switching from depolarization to hyperpolarization during this time in rodent cultures (*Ganguly et al., 2001*; *Rheims et al., 2008*). However, we inhibited $GABA_A$ receptors chronically from DIV 1 and found homophilic rules still provided the best fit. Therefore, whilst $GABA_A$ receptor-mediated inhibition may play a role in de-coupling neurons and hub node function (*Bonifazi et al., 2009*; *Cossart, 2014*; *Mann et al., 2009*), further work is required to understand the precise relationship between inhibition mediated by different receptor subtypes (e.g. $GABA_B$ receptor-mediated), cell types, and how this changes wiring parameter magnitude, and ultimately network topology, over development. Incorporating computational models that sit at the molecular-level (e.g. *Andrews and Iglesias, 2007*; *Libby et al., 2007*; *Mortimer et al., 2009*), as opposed to our network-level models, will be particularly useful in this endeavor. Despite being at a different descriptive level, these models often emphasize highly analogous principles. For example, just as between-neuron spatial relationships are core to how networks form, within-neuron spatial relationships between the center of growth cones and signaled receptors are core to axonal guidance (*Mortimer et al., 2009*). It also remains unclear to what extent the smaller size of gabazine-treated networks, compared to controls, affects other network statistics. However, given the homogeneity of activity across neurons in gabazine-treated networks, smaller network size, rather than flattened probability distributions, is unlikely to explain differences in best-fitting model parameters. Future work should aim to further formalize across-level explanations of neural development.

Another potential limitation relates to the application of our modeling approach to human monolayer and organoid cultures. The human iPSC-derived neuronal networks consist of purified cell-types, which is clearly artificial. As reflected by the lack of connectivity and topology in some of these cultures, such as the DN networks, which also did not show homophilic wiring, there is perhaps little surprise they performed differently. A more veridical account will likely come from cultures with mixed iPSC lines exhibiting more complex spiking dynamics arising from defined cell-type driven heterogeneity (*Perez-Nieves et al., 2021*). It is also possible that the spatial extent of the recording from 3D hCOs onto the 2D HD-MEA system may have limited our network inferences. Despite these caveats, our present work shows that homophilic generative models per se are appropriate growth models for in vitro neuronal networks as they were capable of recapitulating key statistical properties—both at the local and global level. However, as noted in prior GNM studies (*Akarca et al., 2021*; *Oldham et al., 2022*), a significant future advance in this research area will come from weighted generative network models capable of recapitulating weighted topological architectures. Such an approach would allow for both the tuning of connection weights over developmental time—a clear principle of network maturation *Johnson et al., 2019*—but also enable further study of how developing network topology, genetics (*Arnatkeviciute et al., 2021*; *Oldham et al., 2022*; *Oldham and Fornito, 2019*)

and information processing (*Chen et al., 2006*; *Ali et al., 2022*; *Achterberg et al., 2023*) together explain neuronal network organization across scales.

In conclusion, we find that the complex topology of developing rodent and many human neuronal networks in vitro can be simulated via a simple homophily generative model, where neurons aim to maximize locally shared connectivity within an economic context. With this, and prior research at the macroscopic level in mind, we suggest that homophily wiring rules provide a compelling isomorphic explanation for decentralized, locally computing, developing systems.

## Methods
### High-density microelectrode arrays
Two types of CMOS-based high-density microelectrode array (HD-MEA) recording systems, produced by MaxWell Biosystems (Zurich, Switzerland), were used in the present study (*Müller et al., 2015*; *Ballini et al., 2014*). The single-well HD-MEA MaxOne, consists of 26,400 electrodes with a center-to-center electrode pitch of 17.5 μm, arranged in a 120x220 electrode array structure. This HD-MEA can record simultaneously from a total of 1024 user-selected readout-channels at 20 kHz (HD-MEAs have a sensing area of 3.85×2.10 $mm^2$). For more technical details, see previous studies (*Müller et al., 2015*; *Ballini et al., 2014*). The second recording platform was the multi-well HD-MEA MaxTwo system (MaxWell Biosystems), capable of recording from plates with either 6- or 24-wells, that comprised the same number of electrodes and readout-channels and very similar electrode specifications as the MaxOne for each well (while the electrode size is 8.75×12.50 $μm^2$ for the MaxOne chip and HD-MEAs of the 6-well plate, it is 12.0×8.8 $μm^2$ for the 24-well plate HD-MEA). With this system, it is possible to simultaneously record from six wells at a time and at a sampling rate of 10 kHz. To decrease the impedance and to improve the signal-to-noise ratio (SNR), electrodes of single- and six-well HD-MEAs were coated with platinum black (*Müller et al., 2015*).

### Rodent primary cortical neuronal cultures
Before plating, HD-MEAs were sterilized in 70% ethanol for 30 min and rinsed three times with sterile water. To enhance cell adhesion, the electrode area of all HD-MEAs was treated with poly-D-lysine (PDL, 20 μL, 0.1 mg mL$^{-1}$; #A3890401, Gibco, Thermo Fisher Scientific, Waltham, USA) for 1 hr at room temperature and then rinsed three times with sterile water. Next, 10 μL Geltrex (#A1569601, Gibco, 0.16 mg mL$^{-1}$) was pipetted on each array and again left for about 1 hr at room temperature. For the main analysis of the paper, we used rodent primary cortical (PC) neurons prepared as previously described (*Ronchi et al., 2019*). Briefly, cortices of embryonic day (E) 18/19 Wistar rats were dissociated in trypsin with 0.25% EDTA (Gibco, #25200–056), washed after 20 min of digestion in plating medium (see below), and triturated. Following cell counting with a hemocytometer, either 50,000 cells (~1000 cells/$mm^2$; here referred to as 'sparse' plating condition) or 100,000 cells (2000 cells/$mm^2$; referred to as 'dense' plating condition) were seeded on each array and afterwards placed in a cell culture incubator for 30 min at 37 °C/5% $CO_2$. Next, the plating medium was added carefully to each well. The plating medium contained 450 mL Neurobasal (Invitrogen, Carlsbad, CA, United States, #21103049), 50 mL horse serum (HyClone, Thermo Fisher Scientific, #SH30074), 1.25 mL Glutamax (Invitrogen, #35050–038), and 10 mL B-27 (Invitrogen, #A3582801). After 2 days, half of the plating medium was exchanged with growth medium containing 450 mL D-MEM (Invitrogen, #11960–044), 50 mL horse serum (HyClone), 1.25 mL Glutamax (Invitrogen), and 5 mL sodium pyruvate (Invitrogen, #11360–039). Across all experiments, the medium was exchanged twice a week, at least one day before the recording sessions. All animal experiments were approved by the veterinary office of the Kanton Basel-Stadt (license #2358) and carried out according to federal laws on animal welfare. All cell lines were tested negative for mycoplasma contamination. A summary of the data used is provided in *Supplementary file 1a*.

Notably, cell densities were estimated only on the day of plating and not on the actual recording days. As reported in previous research (*Xing et al., 2021*), it is assumed that the actual cell numbers on the HD-MEAs, on the respective recording days, were significantly lower than the initially plated numbers. This reduction may result from incomplete adherence to the HD-MEA, stress during the plating process, the composition of plated cells, and/or naturally occurring cellular processes such as apoptotic cell death.

## Rodent primary cortical culture validation dataset

To probe the reproducibility of our results, we obtained a second set of sparse rodent PC cultures. The protocol to prepare this dataset was very similar to the procedures described previously for our main datasets. The wells of a 24-well HD-MEA plate (MaxWell Biosystems) were sterilized in 70% ethanol for 30 min and rinsed three times with sterile water. To enhance cell adhesion, the electrode area of the HD-MEAs was treated for 1 hr at room temperature with 20 μL of 0.05% (v/v) poly(ethyleneimine) (Sigma-Aldrich, #181978) in borate buffer (Thermo Fisher Scientific, #28341) at 8.5 pH, and then washed three times with sterile water. Next, 10 μL of laminin (0.02 mg ml$^{-1}$; Sigma-Aldrich, #L2020) in Neurobasal medium (Gibco, Thermo Fisher Scientific) was pipetted on each array and again left for 1 hr at room temperature. Rodent PC neurons were prepared as previously described (*Ronchi et al., 2019*). The plating density for this dataset was 50,000 cells per well (~1000 cells/mm²); the recordings included in the replication analysis were performed on DIV 14. The results for this analysis are depicted in *Figure 7—figure supplement 3*.

## Human induced pluripotent stem cell-derived neuronal cultures

Three different human iPSC-derived neuronal cell lines were included in the study: iCell DopaNeurons, iCell Motor Neurons, and iCell GlutaNeurons, all commercially available from FUJIFILM Cellular Dynamics International (FCDI, Madison, USA). All neural cells were co-cultured with human iCell Astrocytes (FCDI, see above). Cell plating: Cell plating medium consisted of 95 mL of BrainPhys Neuronal Medium (STEMCELL Technologies, Vancouver, Canada), 2 mL of iCell Neuronal Supplement B (FCDI), 1 mL iCell Nervous System Supplement (FCDI), 1 mL N-2 Supplement (100 X, Gibco), 0.1 mL laminin (1 mg/mL, Sigma-Aldrich), and 1 mL Penicillin-Streptomycin (100 X, Gibco). Neurons and astrocytes were thawed in a 37 °C water bath for 3 min. The cells were then transferred to 50 mL centrifuge tubes, and 8 mL plating medium (at room temperature) was carefully added. Cell suspensions were centrifuged at 380 × g (1600 RPM) for 5 min, and the supernatant was aspirated. Cell pellets were then resuspended in plating medium and combined to achieve a final concentration of 10,000 neurons and 2000 astrocytes per μL. Finally, 100,000 neurons (10,000 cells/mm²) and 20,000 astrocytes were seeded per HD-MEA by adding 10 μL of the prepared solution, after removing the Geltrex droplet. After incubating the cells for 1 hr at 37 °C/5% CO$_2$, another 0.6 mL (for MaxOne chips of the PSM type)/1.2 mL (for MaxOne chips of the PLM type) of plating medium was added. Half of the medium was changed twice a week. For more details, see *Ronchi et al., 2021*.

## Human cerebral organoid slice cultures

Human embryonic stem cell (ESC)-derived cerebral organoids (hCOs) were generated from a commercially available human ESC line (Takara Bio, Osaka, Japan), using the STEMdiff cerebral organoid kit (STEMCELL Technologies) following the manufacturer's instructions. Slices were obtained from 120-days-old hCOs. Single organoids were first transferred from maturation medium to ice-cold BrainPhys (STEMCELL Technologies) using cut 1000 μl pipette tips. Next, cross-sectional 500-μm-thick slices were cut from hCOs using a sterile razor blade and collected in petri dishes filled with BrainPhys medium at room temperature. Before the plating, HD-MEAs were sterilized in 70% ethanol for 30 min and rinsed three times with distilled water. To improve tissue adhesion, arrays were coated with 0.05% (v/v) poly(ethyleneimine) (Sigma-Aldrich) in borate buffer (pH 8.5, Thermo Fisher Scientific) for 30 min at room temperature, rinsed with distilled water, and left to dry. To attach hCOs on HD-MEAs, we applied a thin layer of Matrigel (Corning) to the center of the HD-MEA and then transferred individual organoid slices to the coated HD-MEAs. After positioning the tissue, we placed a tissue 'harp' on top of the organoid slice and applied several drops of recording medium (see below; STEMCELL Technologies, #05793) around the organoid. HD-MEAs were then covered with a lid and placed in a humidified incubator at 37 °C, 5% CO$_2$/95% air for 30 min, before adding more medium to a final volume of 2 ml per chip. Half of the recording medium was changed every 2–3 days. The recording medium is composed of BrainPhys, N2-A supplement, and SM1 neuronal supplement (for 10 mL of BrainPhys, we added 100 μL of N2 supplement and 200 μL of SM1 supplement). Note that all three forms of neuronal culture described in this section and above (rodent PC networks, human monolayer neuronal cultures, and organoid cultures) have been replicated many times internally in the laboratory.

## Immunohistochemistry

PC neurons were fixed using 4% paraformaldehyde solution (Thermo Fisher, #FB001). Samples were permeabilized and blocked using a PBS 10 X (Thermo Fisher, #AM9625) solution containing 10% normal donkey serum (NDS) (Jackson ImmunoResearch, West Grove, USA, #017000001), 1% bovine serum albumin (BSA) (Sigma-Aldrich, #05482), 0.02% Na-Az (Sigma-Aldrich, #S2002), and 0.5% Triton X (Sigma-Aldrich, #93443). Permeabilization facilitated antigen access to the cell, while blocking prevented non-specific binding of antibodies to neurons. Primary and secondary antibodies were diluted in a PBS solution containing 3% NDS, 1% BSA, 0.02% Na-Az, and 0.5% Triton X. The used antibodies are also listed in *Supplementary file 1i*. Note, immunohistochemistry was performed on control PC cultures prepared as previously outlined (*Ronchi et al., 2021*).

Human iPSC-derived neurons were fixed using 8% PFA solution (#15714 S, Electron Microscopy Sciences) and blocked for 1 hr at room temperature (RT) in blocking buffer containing 10% normal donkey serum (NDS) (Jackson ImmunoResearch, West Grove, USA, #017-000-001), 1% bovine serum albumin (BSA) (#05482, Sigma-Aldrich), and 0.2% Triton X (Sigma-Aldrich, #93443) in PBS (Thermo Fisher Scientific, #AM9625). Primary antibodies (*Supplementary file 1i*) were diluted in a blocking buffer and incubated overnight at 4 °C. Samples were washed three times with 1% BSA in PBS and incubated with the secondary antibody (*Supplementary file 1i*) diluted in blocking buffer for 1 hr at RT. After three additional washes with PBS, DAPI was added for 2 min at RT (1:10000). Images were acquired using the Opera Phenix Plus High-Content Screening System (#HH14001000, PerkinElmer, Waltham, MA, USA).

hCOs were fixed using 4% paraformaldehyde (PFA) for 4 hr at room temperature, washed with PBS, and immersed in 30% sucrose solution at 4 °C overnight. PFA-fixed organoids were embedded in OCT compound (Sakura Finetek, Alphen aan den Rijn, Netherlands, #4583) and stored at –80 °C. 10 µm sections were cut on a cryostat and collected on Superfrost plus slides (Thermo Scientific, #22-037-246). For immunohistochemistry, sections were permeabilized in 0.1% Triton X-100 and blocked with animal-free blocker (Vector Laboratories, Burlingame, CA, USA, #SP-5030–250). Slides were incubated with primary antibodies for 1 hr at room temperature. Sections were washed in PBS and further incubated with secondary antibodies for 1 hr at room temperature. After washing with PBS, sections were incubated with PureBlu DAPI (Bio-Rad, Hercules, CA, USA, #1351303) for 3 min and mounted with ProLong Gold antifade mounting medium (Thermo Scientific, #P36930). Fluorescence images were acquired with a SP8 confocal microscope (Leica, Wetzlar, Germany). The primary and secondary antibodies used for hCO stainings are listed in *Supplementary file 1i*.

## Scanning electron microscope imaging

Fresh tissue samples were fixed in 2.5% glutaraldehyde solution (Sigma-Aldrich, St. Louis, USA) overnight. After fixation, the samples were dehydrated in an ascending acetone series (50%, 70%, 80%, 90%, 95%, 100%), and critically point dried (CPD; Quorum Technologies, West Sussex, UK), using $CO_2$ as the substitution fluid. The procedure is generally suited for SEM preparation and ensures that surface structures of animal tissue samples are preserved in their natural state, i.e., without shrinkage, distortion, or dissolution. After CPD, specimens were carefully mounted on aluminum stubs using double-sticky carbon-coated tabs as adhesive (Plano, Wetzlar, Germany). Thereafter, they were coated with gold-palladium in a sputter device for 45 s (Bio-Rad SC 510, Munich, Germany). SEM analyses were carried out with a Zeiss Digital Scanning Electron Microscope (SUPRA 40 VP, Oberkochen, Germany) in SE2 mode at 5–10 kV.

## Electrophysiological recordings

To characterize rodent in vitro neuronal networks on HD-MEAs, and to track their functional connectivity, we performed developmental recordings, starting one week after the plating. Using the MaxLab Live software (MaxWell Biosystems), we first used the 'Activity Scan Assay' module, which performs a series of high-density recordings, to screen for active electrodes across the entire HD-MEA. During the activity scan the multi-unit activity for each electrode was estimated by applying an online sliding window threshold-crossing spike-detection algorithm. After the activity scan, we used the MaxLab Live 'Network Assay' module to select up to 1024 readout-electrodes based on the identified active electrodes. To track networks at single-cell resolution over development, we obtained high-density network recordings, consisting of up to 64 non-overlapping 4×4 electrode blocks

(electrode-to-electrode pitch of 17.5 µm), and a minimum spacing of at least 87.5 µm between the centroids of the blocks. Depending on the dataset at hand, the block configurations were based on an activity scan performed on DIV 7 (sparse PC networks) or DIV 14 (dense PC networks). The duration of the HD-MEA network recordings was 30 min (for each day); an overview on the different datasets is provided in *Supplementary file 1a*. The sparse/dense rodent PC neuronal networks (main manuscript) and the hCO's were recorded on MaxTwo six-well plates (MaxWell Biosystems); the human iPSC-derived neurons (glutamatergic, motor, and dopaminergic neurons) were recorded on single-well MaxOne HD-MEAs (MaxWell Biosystems; see also *Ronchi et al., 2021*). Finally, the validation dataset (*Figure 7—figure supplement 3*), comprising sparse rodent PC neuronal networks, was acquired on 24-well plates (MaxWell Biosystems).

## Spike-sorting and post-processing

HD-MEA network recordings underwent an initial quality control to assess the overall noise level and signal stability of each recording. Next, we used the software package Kilosort (version 2.0; *Pachitariu et al., 2016*) to spike sort the data, applying default parameters. For the developmental tracking analyses, we concatenated all recordings (i.e. DIV 7, 10, 12, and 14 for the sparse rodent PC cultures, and DIV 14 and 28 for the dense rodent PC cultures). After spike sorting, we inferred array-wide spike-triggered averages (STAs) for all units labeled as 'good' by Kilosort. Next, we calculated the spatial similarity between all detected units/STAs to minimize the influence of potential cluster splits that might have occurred during spike sorting of bursty spontaneous activity. The spatial similarity among the inferred array-wide templates was probed by the normalized pairwise maximum cross-correlation: units/STAs that showed a similarity $r > 0.75$ and had at least five electrodes in common underwent an iterative elimination process using a simple clustering heuristic (based on a standard modularity algorithm; *Blondel et al., 2008*). Please see *Supplementary file 1a* for a summary of the datasets used.

To probe whether the reported main results depend on our spike-sorting/post-processing pipeline, we also prepared the data using an alternative pipeline. This alternative pipeline used a later version of Kilosort (version 2.5), accessed through SpikeInterface (*Buccino et al., 2020*), and the quality control toolbox Bombcell (*Fabre et al., 2023*) to screen for units that may be included in the analysis. Moreover, we restricted functional connectivity inference (see below) to unit pairs with a minimum of co-activity, i.e., at least 50 spikes within a±50 ms time window of the cross-correlogram calculated between the units. We also excluded neuron pairs with potential spike sorting issues, slightly modifying the spike sorting index introduced by a recent study to our data (*Ren et al., 2020*). As we show in *Figure 7—figure supplement 2*, results from this analysis were mostly in line with the initial results reported in the main manuscript. Of note, for most datasets, the number of included units was higher with the alternative sorting/post-processing pipeline (*Figure 1—figure supplement 1*).

## Pharmacological experiments

Pharmacological experiments with the $GABA_A$ receptor blocker gabazine (SR 95531 hydrobromide, Sigma-Aldrich, #104104509), were performed on sparse rodent PC neuronal cultures. Nine cultures were treated with 1 µM gabazine one day after plating and tracked until DIV 14; media + gabazine media exchanges were performed 2–3 times per week.

## Firing rate and burst statistics

Firing rates across neuronal units were calculated as the total number of spikes per unit time (in seconds) in the entire recording. Array values were calculated as the mean across all active units (firing rates >0.01 Hz). Burst rates were calculated using a maximum interspike interval (ISI) method (*Cotterill et al., 2016*) based on the ISI between every N-th spike ($ISI_N$; *Bakkum et al., 2013b*). The $ISI_N$ threshold for determining the onset/offset of bursting activity was determined by finding the local trough in the bimodal logISI distribution (see *Figure 6—figure supplement 1b*). The two peaks, at short ISIs and long ISIs represent more high frequency bursting and regular activity, respectively. The coefficient of variation (CV) of interburst intervals (IBIs) was calculated as the standard deviation of IBIs relative to the mean IBI of a given neuronal unit; the array value was the mean of this across all neuronal units.

## Functional connectivity inference

To detect pairwise correlations in spike trains, here referred to as functional connectivity, we computed the spike time tiling coefficient (STTC; *Cutts and Eglen, 2014*). The STTC aims to mitigate potential confounding in basic correlation indices introduced by different firing rates, by quantifying the proportion of spikes in one train which fall within ± Δt (the synchronicity window) of another. It is given by:

$$STTC = \frac{1}{2}\left(\frac{P_A - T_B}{1 - P_A T_B} + \frac{P_B - T_A}{1 - P_B T_A}\right), \tag{4}$$

where $T_A$ is the proportion of total recording time which lies within $\pm\Delta t$ of any spike from A ($T_B$ is calculated similarly). $P_A$ is the proportion of spikes from A which lies within $\pm\Delta t$ of any spike from B ($P_B$ is calculated similarly). The synchronicity window, $\Delta t$, is the only free parameter in the STTC calculation. In the present study, we used a $\Delta t=10$ ms. A visualization of the STTC calculation is provided in *Figure 1h*; STTC was calculated using publicly available Matlab code (*Cutts and Eglen, 2014*). We used permutation-based testing to determine the significance of connections: For a given neuronal unit's spike train, spike times were randomly jittered by sampling from a normal distribution with a standard deviation of 10 ms, generating a surrogate spike train. The code for jittering spike trains was adopted from the Neural Complexity and Criticality Toolbox (*Marshall et al., 2016*). The jittering and STTC inference procedure was repeated for each neuronal unit for 1000 permutations. To calculate the significance of pairwise functional connectivity, experimentally inferred STTC values were compared to the distribution of surrogate SSTC values. A significance value of p<0.001 was used as a cutoff to binarize functional connectivity matrices and to calculate network related analysis throughout the manuscript; only units with firing rates >0.01 Hz were considered.

## Transfer Entropy Estimation

We used the CoTETE package (https://github.com/dpshorten/CoTETE.jl; *Shorten, 2022*) to infer transfer entropy (TE). CoTETE allows for TE estimation between even-based time series in continuous time, i.e. without binning of the data (*Shorten et al., 2021*). Parameters that were changed from their defaults are listed in *Table 1* and were largely adapted from *Shorten et al., 2021*. For each putative connection, the distribution of surrogate TE values was approximated by a normal distribution with mean and variance estimated from the surrogates. The corresponding cumulative distribution function (CDF) was then evaluated at the empirical TE value. The p-value was defined as the probability of observing a surrogate TE greater than or equal to the empirical TE, i.e., $p=1-CDF(TE_{emp})$.

## Network statistics

In *Figure 2a*, we provide a visualization of key graph theoretical metrics relevant for this study. Here, we provide both a written and mathematical definition for each measure used. Each statistic was calculated using the Brain Connectivity Toolbox (*Rubinov and Sporns, 2010*):

**Table 1.** Non-default parameter values used for the continuous-time TE inference.

| Parameter | Description | Value |
|---|---|---|
| IX | Number of interspike intervals in target history embeddings | 4 |
| IY | Number of interspike intervals in source history embeddings | 2 |
| kglobal | Number of nearest neighbours to find in the initial search | 10 |
| kperm | Number of nearest neighbours to consider during surrogate generation | 4 |
| Nsurrogates | Number of surrogates to generate for each node pair | 100 |
| sampling_method | Method with which to place the random sampling points. | "jittered_target" |
| jittered_sampling_noise | Width of the uniform jitter added to the target spike times used in resampling when sampling_method is set to "jittered_target". | 200.0 |

## Network size and total degree

The network size here refers to the number of *nodes* in the network. This excludes nodes that did not form any connections (neuronal units with <0.01 spikes/s were already excluded as above). Network size is sometimes referred to as network order in graph theory literature to avoid confusion with the number of *edges*, however, we refer to the total number of edges in the network as total degree.

## Degree

The degree is the number of edges connected to a node. The degree of node *i* is given by:

$$k_i = \sum_{j \in N} a_{i,j},$$

(5)

where $a_{i,j}$ is the connection status between *i* and *j*. $a_{i,j}$ = 1 when link *i,j* exists (when *i* and *j* are neighbors); $a_{i,j}$ = 0 otherwise ($a_{i,i}$=0 for all *i*).

## Clustering coefficient

The clustering coefficient is the fraction of a node's neighbors that are neighbors of each other. The clustering coefficient for node *i* is given by:

$$c_i = \frac{1}{n} \sum_{i \in N} \frac{2t_i}{k_i (k_i - 1)},$$

(6)

where $c_i$ is the clustering coefficient of node *i* ($c_i$=0 for $k_i < 2$).

## Betweenness centrality

The betweenness centrality is the fraction of all shortest paths in the network that contain a given node. Nodes with high values of betweenness centrality, therefore, participate in a large number of shortest paths. The betweenness centrality for node *i* is given by:

$$b_i = \sum_{h,j \in N} \frac{\rho_{hj}(i)}{\rho_{hj}},$$

(7)

where $\rho_{hj}$ is the number of shortest paths between *h* and *j*, and $\rho_{hj}(i)$ is the number of shortest paths between *h* and *j* that pass through *i*.

## Edge length

The edge length is the total edge length connected to a node. It is given by:

$$d_i = \sum_{j \in N} a_{i,j} d_{i,j},$$

(8)

where $d_{i,j}$ is the Euclidean distance between *i* and *j*. The Euclidean distances of functional connectivity graphs inferred in the present study are depicted in ***Figure 2—figure supplement 1***.

## Global efficiency

The global efficiency is the average of the inverse shortest path length. It is given by:

$$E_i = \frac{1}{n} \sum_{i \in N} \frac{\sum_{j \in N, j \neq i} d_{i,j}^{-1}}{n - 1},$$

(9)

where here, $d_{i,j}$ represents the shortest path length between *i* and *j*. $d_{i,j}^{-1}$, therefore, represents the inverse shortest path length, meaning that a high efficiency corresponds to a low shortest path length.

## Local efficiency

The local efficiency of the network is given by:

$$E_{\text{loc}} = \frac{1}{n} \sum_{i \in N} E_{\text{loc},i} = \frac{1}{n} \sum_{i \in N} \frac{\sum_{j,h \in Nj \neq i} a_{ij}, a_{ih} \left[ d_{jh}(N_i) \right]^{-1}}{k_i (k_i - 1)} \qquad (10)$$

where $E_{loc}$ is the local efficiency of node $i$, and $d_{jh}(N_i)$ is the length of the shortest path between $j$ and $h$, that contains only neighbors of $i$.

## Matching

The matching index computes the proportion of overlap in the connectivity between two nodes. It is given by:

$$M_{i,j} = \frac{\left| N_{i/j} \cap N_{j/i} \right|}{\left| N_{i/j} \cup N_{j/i} \right|}, \qquad (11)$$

where $N_{i/j}$ refers to the neighbors of the node $i$ excluding node $j$. Where global measures of matching have been used, we averaged across the upper triangle of the computed matching matrix.

## Small-worldness

Small-worldness refers to a graph property where most nodes are not neighbors of one another, but the neighbors of nodes are likely to be neighbors of each other. This means that most nodes can be reached from every other node in a small number of steps. It is given by:

$$\sigma = \frac{c/c_{rand}}{l/l_{rand}}, \qquad (12)$$

where $c$ and $c_{rand}$ are the clustering coefficients, and $l$ and $l_{rand}$ are the characteristic path lengths of the respective tested network and a random network with the same size and density of the empirical network. Networks are generally considered as small-world networks at σ>1. In our work, we computed the random network as the mean statistic across a distribution of n=1000 random networks. The characteristic path length is given by:

$$L_i = \frac{1}{n} \sum_{i \in N} \frac{\sum_{j \in N, j \neq i} d_{i,j}}{n - 1}, \qquad (13)$$

## Modularity

The modularity statistic, Q, quantifies the extent to which the network can be subdivided into clearly delineated groups:

$$Q = \frac{1}{l} \sum_{i,j \in N} \left( a_{i,j} - \frac{k_i k_j}{l} \right) \delta_{m_i m_j}, \qquad (14)$$

where $m_i$ is the module containing node $i$, and $\delta_{m_i m_j} = 1$ if $m_i = m_j$, and 0 otherwise.

## Participation coefficient

The participation coefficient is a measure of diversity of intermodular connections of individual nodes, where community allocation was determined via a Louvain algorithm, with a resolution parameter $\gamma$=1, which aims to form a subdivision of the network which maximizes the number of within-group edges and minimizes between-group edges.

## Generative network modeling

The generative network model can be expressed as a simple wiring equation, where wiring probabilities are computed iteratively by trading-off the cost of forming a connection, against the value of the connection being formed in terms of a network topology term. Connections are added iteratively according to these wiring probabilities. It is given by the wiring equation as provided in *Equation 1*. The $D_{i,j}$ term represents the 'costs' incurred between neuron modeled as the Euclidean distance between tracked units (*Figure 2—figure supplement 1*). The $K_{i,j}$ term represents how neurons 'value'

each other, given by an arbitrary topological relationship which is postulated a priori (also termed 'wiring rule' given mathematically in *Supplementary file 1b*). $P_{i,j}$ reflects the probability score of forming a fixed binary connection at the current time step. Note that, as this is a probability score, the sum of the values may not necessarily equal one. This is because the score indicates the relative likelihood of nodes forming a connection over others. The simulation continues until the simulated network has the same number of connections as the observed network. The $D_{i,j}$ term remains constant during the simulation, while the $K_{i,j}$ term updates at each time point (and, therefore, also the $P_{i,j}$ term).

## Cost functions

In the present study, we make a distinction between simulated networks which mirror the statistical distributions of observed networks and those which mirror the topological organization of those statistics. The former can be accessed via a previously established energy equation (*Betzel et al., 2016*), whereby the model fit is given by the 'worst' of the four *KS* distances assessed, given by *Equation 2*. *KS* is the Kolmogorov-Smirnov statistic between the observed and simulated networks at the particular $\eta$ and γ combination used to construct the simulation, defined by the network degree *k*, clustering coefficient *c*, betweenness centrality *b*, and Euclidean edge length *d*. Notably, the KS distance between two vectors simply considers their statistical *distributions*. When we report model fits in the main report, we group the model fits based on the broad generative rule group (as shown by the distinct colors in *Figure 2d*). *Supplementary file 1c* contains a summary of all optimized parameters and model fits. *Supplementary file 1d-h* show all statistical comparisons (One-way ANOVA and pairwise comparisons) between generative rule groups, over all time points for each considered analysis. This was done to simplify the main report, and whenever we report grouped results, we provide Figure supplements which show all results broken down by the individual model.

In *Figure 3—figure supplement 5*, we further assess the ability of the best performing generative models in each class (spatial, matching, clustering average, and degree average) to recapitulate network statistics as included in the energy equation, but also two measures outside (local efficiency and participation coefficient). We did this via a Monte Carlo sampling procedure (*Beasley and Rodgers, 2012*). First, we took the top n=99 performing simulations for each sparse rodent culture's model considered and computed each of the six local statistics as shown in *Figure 3—figure supplement 5* as cumulative density plots. For each statistic, we computed a KS statistic between the observed local statistics distribution and an average of the statistics of the 99 simulations. We then undertook 99 individual leave-one-out iterations in which we replaced a single simulation of the 99 with the observed distribution. For each of the 99 permutations, we computed the same statistic, forming a null distribution. We then calculated a $p_{rank}$ by ranking how close the original observed statistic was to the mean of this computed null distribution (i.e. how close was the observation to the middle of the null). This was computed for each culture and statistic, for each of the considered generative models. We then quoted the median $p_{rank}$ across cultures.

Later in the study, we provide an alternative but simple cost function which does not assess distributions of statistics but instead assesses the *topological fingerprint dissimilarity* of these network statistics. The *TF* matrix is calculated as a Pearson's correlation matrix between each pair-wise combination of the local statistics. In our study, we used six common network statistics to form this correlation matrix; however, in principle, these can be extended to any number or range of local statistical measures. Moreover, there are alternative procedures to the one provided that could be taken to estimate the dissimilarity between the topological fingerprint matrices, that may or may not provide greater capacity to distinguish generative model performances. For example, the Pearson's correlation (as in representational similarity analysis; *Kriegeskorte et al., 2008*) or the non-Euclidean geodesic distance (which better captures the non-Euclidean geometry of functional connectivity in functional magnetic resonance imaging; *Venkatesh et al., 2020*) could be used. Future work should examine the advantages and disadvantages of these approaches for the case of generative network models.

The construction of the *TF* is visualized in *Figure 4a*. The $TF_{dissimilarity}$ is then calculated as the Euclidean norm (*Lindfield and Penny, 2019*) of the difference between the observed and simulated *TF* matrices. This is given in *Equation 3*.

## Parameter selection

We optimized $\eta$ and γ using a Voronoi tessellation procedure as used in prior work (**Betzel et al., 2016**). This procedure works by first randomly sampling the parameter space and evaluating the model fits of the resulting simulated networks, via the energy equation. As there is little prior literature that can be used to guide the present study, we considered a wider range of parameter values, with $\eta$ values in the range from –10 to 10 and γ values in the range –10 to 10. Following an initial search of 4000 parameters in this space, we then applied a Voronoi tessellation to partition the space into two-dimensional cells. We then preferentially sampled from cells with better model fits according to the energy equation (see **Betzel et al., 2016** for further detail). Preference was computed with a severity of $\alpha=2$, which determines the extent to which cell performance led to preferential sampling in the next step. This procedure was repeated a further four times, leading to a total of 20,000 simulations being run for each considered network across the 13 generative rules, as described in **Supplementary file 1b**.

Our search was intentionally extensive, such that we had a greater likelihood of converging on accurate parameter estimates for each culture with the capacity to generate good model fits. In the main paper, we report the top-performing n=1 parameter simulation, but in each case, we replicate our findings across a variable number of high-performing parameters (n=10 and n=50). This was done to show robustness to possible stochasticity entering the model, possibly biasing the obtained results. Of note, recent work has provided a new solution to enable easier convergence on true parameter estimates in generative models (**Liu et al., 2023**), but this approach requires the sample to have the same geometric structure, which is not the case across our HD-MEA recordings.

Due to different amounts of data across datasets (e.g. see **Supplementary file 1c**), we conducted a subsampling procedure in which we sampled from our largest dataset (dense rodent PC cultures) to assess to what extent results may depend on the dataset size. To do this, we took the n=12 cultures across developmental time points (DIV 14 and DIV 28) and sampled n=6 and n=3 (respectively) samples 1000 times. For each of the 1000 sub-samples, we computed statistical testing on the model energy acquired from all assessed generative models and provided 95% confidence intervals on pairwise comparisons and effect sizes. We report these findings in **Supplementary file 1j**, showing that even when sampling as low as n=3, our results appear to generalize very well, with homophily remaining as the rule generating the lowest energy. Of note, we find a general trend that lowering the empirical sample size increases the p-value, which can be considered by those conducting future studies.

## Generative probability distributions

In **Figure 6d**, we show the mean probability score ($P_{i,j}$) distributions within the generative models fit to gabazine and control networks. This was calculated by measuring the $P_{i,j}$ across all node pairs $i$ and $j$ in the network, in 1% intervals, before plotting the average distribution of $P_{i,j}$ across these timesteps. In **Figure 6—figure supplement 4**, we show each distribution of these probability distributions (that was averaged to provide comparisons in **Figure 6d**).

Note that the probability score distribution 'flattening' (from a relative lognormal distribution toward a uniform distribution) means that relatively more candidate connections come to have a higher probability of being connected in the future. This leads to a decreased specificity of future wiring (i.e. the scope of possible outcomes increases). This flattening effect is equivalent to the network outcomes becoming more random, because a greater number of possible future connections have a non-trivial probability score, and the formation of a connection is less driven by the underlying network topology. Note, a completely singular and uniform $P_{i,j}$ distribution (i.e. where all connections have the same probability of wiring) generates an entirely random graph (**Erdős and Rényi, 1959**). In contrast, when a generative network has a heterogeneous distribution in its $P_{i,j}$ distribution (e.g. with large numbers of small probability scores and small numbers of large probability scores), the network will have a smaller scope for variability. The probability score distributions, $P_{i,j}$, shown in this work, are plotted with the score value on the x-axis and the frequency on the y-axis. It is of note that there are alternative ways that one could represent these distributions, such as the edge index on the x-axis and $P_{i,j}$ on the y-axis, which would provide different appearing distributions (for more detail, see **Carozza et al., 2023a**; **Abril et al., 2018** which provides a detailed analysis of the magnitude of randomness of generated networks under different parameter conditions).

## Autocorrelogram analysis

Autocorrelogram analysis (as in *Figure 7—figure supplement 1*) was carried out using the CCG function provided by the CellExplorer Matlab Toolbox (*English et al., 2017*; *Petersen et al., 2021*). First, spike times were concatenated cumulatively across units to give a single vector of spike times. Spikes were summed into consecutive one-millisecond bins giving a vector where each element is a one-millisecond bin containing the number of spikes occurring in the network at each time point. This vector, $v$, was then correlated with itself plus a lag value, $x$. The range of lag values tested was between –500 and 500 ms. Lag values between –1–1ms were removed to impose a refractory period—hence, these values are 0 in *Figure 7—figure supplement 1*. For example, where $x$ is 5ms, the CCG function is the sum of $v_i$ and $v_{t+x}$ across all time points, $t$. This gives a vector of CCG values, corresponding to spatiotemporal overlap in spike times in increment 1 ms 1ms. To control for variability in firing rate between recordings, the CCG values were normalized to the maximum value in this CCG vector.

## Code availability

Results were generated using code written in Matlab 2020b. All code is available at (https://github.com/DanAkarca/MEA_generative_models, copy archived at *Akarca, 2025*).

## Acknowledgements

This work was supported by the European Union through the European Research Council (ERC) Advanced Grant 694829 'neuroXscales' and the corresponding proof-of-concept Grant 875609 'HD-Neu-Screen,' by the two Cantons of Basel through a Personalized Medicine project (PMB-01–18), granted by ETH Zurich, the Innosuisse Project 25933.2 PFLS-LS, the Swiss National Science Foundation under contract 205320_188910/1 and a Swiss Data Science Center project grant (C18-10). Danyal Akarca and Alexander Dunn are supported by the Medical Research Council Doctoral Training Programme. Danyal Akarca is supported by the Cambridge Trust Vice Chancellor's Award Scholarship. Duncan Astle is supported by Medical Research Council Program Grant MC-A0606-5PQ41. Both Duncan Astle and Danyal Akarca are supported by The James S McDonnell Foundation Opportunity Award. Congwei Wang and Marco Terrigno are supported by Roche postdoctoral fellowship program. Petra Vertes is a fellow of MQ: Transforming Mental Health (MQF17_24). We thank Dr Martin Oeggerli for contributing the serial section electron microscopy image (*Figure 1a*), and the IT department at the MRC Cognition and Brain Sciences Unit, Cambridge, as well as the HPC team at ETH Zürich, for assistance with high performance computing.

## Additional information

### Competing interests

Silvia Ronchi: is employed by MaxWell Biosystems AG, which commercializes HD-MEA technology. Congwei Wang, Marco Terrigno, Ravi Jagasia: is affiliated with F Hoffmann-La Roche Ltd The author has no financial interests to declare. The other authors declare that no competing interests exist.

### Funding

| Funder | Grant reference number | Author |
|---|---|---|
| European Research Council | 10.3030/694829 | Philipp J Hornauer<br>Andreas Hierlemann<br>Manuel Schröter |
| Medical Research Council | MC-A0606-5PQ41 | Danyal Akarca<br>Alexander WE Dunn<br>Duncan E Astle |
| James S. McDonnel Foundation | | Danyal Akarca<br>Duncan E Astle |
| MQ: Transforming Mental Health | MQF17_24 | Petra E Vértes |

| Funder | Grant reference number | Author |
|---|---|---|
| European Research Council | 10.3030/875609 | Philipp J Hornauer Andreas Hierlemann Manuel Schröter |
| ETH Zurich | PMB-01–18 | Andreas Hierlemann |
| Swiss National Science Foundation | 205320_188910/1 | Manuel Schröter Andreas Hierlemann |
| Swiss Data Science Center | C18-10 | Manuel Schröter Andreas Hierlemann |

The funders had no role in study design, data collection and interpretation, or the decision to submit the work for publication.

## Author contributions

Danyal Akarca, Alexander WE Dunn, Conceptualization, Resources, Formal analysis, Investigation, Visualization, Methodology, Writing – original draft, Project administration, Writing – review and editing; Philipp J Hornauer, Data curation, Investigation, Methodology; Silvia Ronchi, Michele Fiscella, Congwei Wang, Marco Terrigno, Ravi Jagasia, Data curation, Investigation, Visualization; Petra E Vértes, Investigation, Methodology, Writing – review and editing; Susanna B Mierau, Ole Paulsen, Investigation, Writing – review and editing; Stephen J Eglen, Investigation, Methodology; Andreas Hierlemann, Data curation, Funding acquisition, Investigation, Methodology; Duncan E Astle, Conceptualization, Resources, Supervision, Funding acquisition, Investigation, Methodology, Writing – original draft, Project administration, Writing – review and editing; Manuel Schröter, Conceptualization, Resources, Data curation, Supervision, Funding acquisition, Investigation, Visualization, Writing – original draft, Project administration, Writing – review and editing

## Author ORCIDs

Danyal Akarca (ID) https://orcid.org/0000-0002-5931-0295
Alexander WE Dunn (ID) https://orcid.org/0000-0003-1504-499X
Philipp J Hornauer (ID) https://orcid.org/0000-0003-2265-6679
Petra E Vértes (ID) https://orcid.org/0000-0002-0992-3210
Ole Paulsen (ID) https://orcid.org/0000-0002-2258-5455
Stephen J Eglen (ID) https://orcid.org/0000-0001-8607-8025
Andreas Hierlemann (ID) https://orcid.org/0000-0002-3838-2468
Manuel Schröter (ID) https://orcid.org/0000-0002-9347-9203

## Ethics

All animal experiments were approved by the veterinary office of the Kanton Basel-Stadt (license #2358) and carried out according to Swiss federal laws on animal welfare.

## Decision letter and Author response

Decision letter https://doi.org/10.7554/eLife.85300.sa1
Author response https://doi.org/10.7554/eLife.85300.sa2

# Additional files

## Supplementary files

Supplementary file 1. Supplementary tables. (**a**) Overview of the datasets used in the study. (**b**) A list of all the value $K_{i,j}$ terms that were included in the generative modeling, as given in the wiring equation. (**c**) A summary of all optimized parameters and energy values, for each dataset, across generative rules. (**d**) Statistical comparisons of the sparse rodent culture energy values across generative rules. (**e**) Statistical comparisons of the sparse rodent culture topological fingerprint dissimilarity across generative rules. (**f**) Statistical comparisons of the dense rodent culture topological fingerprint dissimilarity across generative rules. (**g**) Statistical comparisons of the dense rodent culture energy values across generative rules. (**h**) Statistical comparisons of human monolayer neuron and cerebral organoid culture energy values across generative rules. (**i**) Overview of used antibodies. (**j**) Confidence intervals of model energies, generated by subsampling from the dense rodent networks.

MDAR checklist

## Data availability

All data used in this study, along with documentation detailing each dataset, is openly available at https://doi.org/10.5281/zenodo.6109413.

The following dataset was generated:

| Author(s) | Year | Dataset title | Dataset URL | Database and Identifier |
| --- | --- | --- | --- | --- |
| Akarca D, Dunn A, Hornauer P, Ronchi S, Fiscella M, Wang C, Terrigno M, Jagasia R, Vértes P, Mierau S, Paulsen O, Eglen S, Hierlemann A, Astle D, Schröter M | 2024 | Homophilic wiring principles underpin neuronal network topology in vitro | https://doi.org/10.5281/zenodo.6109413 | Zenodo, 10.5281/zenodo.6109413 |

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
