## [Editor Report]

This valuable study examines the principles according to which neurons connect to each other in vitro. The authors show solid evidence that data could be best explained by the homophillic wiring principle where neurons preferentially connect to neurons within overlapping groups.

---

## [Decision Letter]

**Decision letter after peer review:**

Thank you for submitting your article "Homophilic wiring principles underpin neuronal network topology in vitro" for consideration by *eLife*. Your article has been reviewed by 2 peer reviewers, and the evaluation has been overseen by a Reviewing Editor and Panayiota Poirazi as the Senior Editor. The reviewers have opted to remain anonymous.

*Essential Revisions:*

Both Reviewers appreciate the scope and importance of the study in both experimental and modelling components. There was also a consensus among Reviewers that it is necessary to

1) clarify the separation between functional and structural connectivity;

2) provide the actual optimized model parameters in addition to the rough overview in plots like Figure 3C;

3) clarify differences between "more structured" vs. "more random" networks

4) make an effort to control for different amounts of data across datasets. For example, for a larger dataset, analyses could be carried out on a subset of the data to show how results might depend on the dataset size.

*Reviewer #1 (Recommendations for the authors):*

First a list of apparent inconsistencies and minor points of the presentation:

Figure 2b: Is reported to be from network recordings, but seems to have only ~28 nodes based on the histograms. That is lower than reported in S1.

Figure 2b: Why are values for betweenness > 100? Would not expect that, given the equation in the methods. Looks more like the total number of shortest paths is reported, rather than a fraction.

Figure 2d: The list puts models 3-7 as degree-based, and 8-12 as clustering-based; the boxplot has it inverse.

Figure 3f: Maybe more of a question: Why does DIV 7 correspond to 50% of "simulated time"? According to Supp Figure S2, DIV 14 has a median density of around 7% and DIV7 has < 1%. Is simulated time not the fraction of edges placed?

Figure 5: What exactly is "matching" that is depicted in the 6th row and column of the correlation matrices? It is explained in the methods as a measurement related to pairs of nodes, but I would expect a per-node-level measurement.

Figure 6: Aren't the P_ij values a bit high? The way I understand the generative algorithm after reading this manuscript and Akarca et al., 2021, is at each step a single edge is selected and placed according to P_ij. So the value should add up to 1.0. Even if more than one is selected at a time, with a median value around 0.2, we would reach an edge density of 0.2 in a single step, so that can't be it.

Figure 7: Caption for panels b and c swapped

Line 592: The case of "no principle being implemented" is listed as being indicated by high energy for all models. Yet, random null models in S5b all have very low energy. (As expected, just set both wiring parameters to 0.)

Line 1198: The explanation of global efficiency as depicted uses Euclidean distance (d_ij) and is independent of the graph structure.

Global efficiency is explained in the methods, but not the local efficiency that is used in Supp Figure S6.

Supp Figure S11b: I am not sure I understand the plot. the x-axis is labeled "log10", but the tick labels are also logarithmically spaced. Also, is the unit really in ms?

General notes:

Overall, a number of powerful and interesting methods and analyses are employed. But their selection seems a bit random. Between Figure S6 and S9 "participation" is swapped for "matching". The "topological fingerprint" is used to evaluate model match for sparse and dense cultures, but not for the GABA block case. In the GABA block case, the internals of the generative model is inspected, but not for dense vs sparse. And more. This makes the whole work seem disconnected.

I think Figure 5 belongs before Figure 4 as it provides further data supporting the idea that the "matching" algorithm builds networks comparable to the in vitro data. Branching off to the case of denser cultures is separate from that.

If I understood correctly that different wiring parameters (η and γ) are optimized for the same model but different cultures of the same type, then it should be attempted to explain the differences. (Not just for the GABA block case.) Do they depend on the size of the model? Age? Density? Average activity level?

I am curious why the main point of the paper is based on the sparse cultures and the dense cultures are much less prominently featured, although the sparse case has n=6 and the dense case n=12.

Overall, I am surprised by which data are shown in the main figures, and which ones are supplementary.

*Reviewer #2 (Recommendations for the authors):*

General:

I am a little confused about how the statistics of the different generative networks are computed. Sometimes it mentions "top performing simulated networks", sometimes "top 10" or "top 50", others "top n=1", and "top n=99 and average". I do not understand why several results are based on the "top n=1 performing simulated networks". These are probabilistic models for small networks, how is the best simulation representative of the underlying wiring principle? The original methodological paper (Betzel, Neuroimage 2016) appears to report the top 1% of energies (100 networks).

Similarly, it is often difficult to understand how significance was assessed across many figures and tables. For example for the anovas, and cohens effects on the energies. Sometimes energies are presented with 4 groups, others with 13. It is unclear if the same procedure was always followed.

Line-based:

159: It is very difficult to relate the reported plating densities with densities reported in other studies. It'd be much better to report real densities (in cells/ mm2) measured at the time of recording (or after staining). See for example the works by Potter, Segal, Moses, labs, and others.

171: After the whole-array activity scan, how were the 4x4 blocks selected? I assume one ends up with 64 independent 4x4 blocks, was the distance between them consistent? Was the ranking procedure from 1103 also used? Given that each 4x4 block can identify more than 1 cell, it is unclear why the number of tracked units in the sparse and dense cultures is so similar. Could it be that the electrode selection procedure was biased?

190: Regarding the jittering procedure, it is unclear whether it involved a random shift of either + or – 10 ms? or any value within that range (uniform distribution?)

310: Based on all the energy/exponent maps (e.g., Figure 2d right), the relevant range of exponents is always between -1 and 1 (or -3,3 at best). It is very difficult to read the value of the exponents based on the actual maps (e.g., Figure 3c). Maybe the authors could report the actual exponents somewhere (as they did in Sup Figure 14c for example). These values might help the interpretability of the results (for example, whether the exponents are positive or negative could be quite relevant).

357: What "large topological changes" do the authors refer to? Figure S2 only reports an overall increase in the number of connections. Also, is the network density measured as "total degree / (1/2*N*(N-1))" ? Or is there an extra factor of 2 somewhere? I'm trying to relate the values in S2a S2c, and supp Table 1.

374: Regarding "homophily performs best when approximating empirical networks, but not randomized networks", how did the authors measure that? The full random case in S5b has the lowest energy of all, and lower energy would mean a better fit if I'm not mistaken. Or were the authors only talking about the "randmio" procedure? If so, the authors might want to plot S5b in the same range as the figure it has to be compared with (0 to 0.6).

375: I don't understand how the staining panel relates to the rest of this figure.

376: F3C, to which of the 13 generative models does this plot correspond to? If I'm not mistaken, the exponents are calculated independently for each of the 13 models and 4 time points.

415: Unsure what the R2, r and p measures mean. Is this just pearson on F3e,f right panels, and its p-value?

442: Same as the comment on L159, the authors should report the density at the time of recording. It is also worth noting that there are many more processes affecting the final density than just apoptosis.

444: I'd like to point out a few important references on this matter. Ivenshitz and Segal, J Neurophysiol 2010. Cohen et al., Brain Research 2008. They looked in detail at how different densities affect activity and network connectivity in cultures.

455: Related to L171. I have difficulties understanding the relationship between plating densities, the number of recorded cells, the distance between cells, and inferred connectivity. From S3, the distribution of Euclidean distances for the two plated densities looks very similar. And the number of tracked units in S1 is also very similar. However, S7 reports functional connectivity with smaller edge lengths and higher clustering. This would be consistent if the higher plating density just resulted in more "cells/mm2" and the activity being invariant. But that doesn't seem the case. The authors should clarify that.

509: Why did the authors choose a dissimilarity measure based on the Euclidean norm? Distances between correlation matrices are non-euclidean (see for example Venkatesh et al., Neuroimage 2020).

625: If I understood this measure correctly, the STP is asymmetric STP(i->j) != STP(j->i). If that's the case, did the putative inhibitory connections cluster around specific input cells? i.e., following Dale's rule? That would be an interesting check.

904: How was this replication of the results with TE-based methods quantified? S5b only shows a similar trend as those in Figure 3, but no quantification. Was this measured on the sparse networks only? And were the properties of the TE-based network and the STTC ones similar regarding their topological properties and fingerprint?

1119: How many cultures were treated with gabazine? Here it mentions "Three". But F6 and the text mention n=9 (6 of them with washout).

1172: A significance value of p < 0.01 was chosen for the connectivity. If I understood it correctly, that means, that on average at this threshold, there's a 0.01 probability the observed SSTC value comes from the surrogate distribution. That would result in a 1% network density just from the false positive rate. That's in line with the values reported in S2a at 7 DIV where the network is probably not yet formed. But does that mean that at 14 DIV the expected number of false positives in the inferred connectivity is ~ 20%? Are the inferred networks robust to changes in the binarization threshold?

Figures:

Figure 2d. Energy. The Clustering and Degree labels might have been swapped.

Figure 3c. Is this for a representative network? Or averaged across the networks?

Figure 6d. How is the shift in the probability distribution related to the wiring becoming more random across all timepoints? Not sure what the authors mean by this.

Figure 6d. There's a mention of SF13e,f, which doesn't exist. Could it be S14c,d?

Figure 7c. Why does the tsne plot only include the dense DIV 28 primary cultures? The other datapoints should also be included, e.g., sparse DIV 14 and gabazine experiments.

Supplementary:

Figure 3. left/right panels should be top/bottom panels.

Figure 10. This figure mentions the dense cultures at DIV 14. Why do they report n=6 when dense cultures are n=12? Or is this figure for sparse cultures instead? The caption also mentions "Distributions are plotted for each […]." There are no distributions in this figure.

Figure 11. It would be beneficial to also add a representative raster (panel a) for the washout case. Results in c report a very high firing rate but no bursting rate for the washout case. What does that look like in the raster?

Figure 11b. The axis should read ISI (it's already in the log scale). Caption mentions a tri-modal distribution for gabazine. Very hard to see when the washout distribution is on top.

Figure 12c. Why is the network size so different between controls, gabazine, and washouts? This is never explained, yet several other measures might depend on this value, e.g., total degree.

Figure 15. Why were only the n=12 DIV 14 dense cultures included here?

[Editors' note: further revisions were suggested prior to acceptance, as described below.]

Thank you for resubmitting your work entitled "Homophilic wiring principles underpin neuronal network topology in vitro" for further consideration by *eLife*. Your revised article has been evaluated by Panayiota Poirazi (Senior Editor) and a Reviewing Editor.

The manuscript has been improved but there are some remaining issues that need to be addressed, as outlined below:

*Reviewer #2 (Recommendations for the authors):*

The authors have addressed the points I listed in my original recommendations for the authors. I thank them for the effort.

With respect to my point about "random" vs. "specific" wiring in the context of the GABA block I now have an additional concern:

I now understand that the Pi,j values are not direct probabilities, but just indicate relative preferences for edge placement. In that case, I am not sure a direct comparison of distributions in Figure 6d, and of their means is very meaningful. If all values for Pi,j of a given model were multiplied by a constant >1 then it would not affect the model wiring, as the relative preferences remained unchanged. But in a plot such as 6d it would stretch the distribution out and increase its mean.

Besides, after reading the authors' reply where they elaborate on 6d, I now wonder if there is some confusion about the following two potential ways to plot Pi,j: First, where all potential edges are along the x-axis and Pi,j along the y-axis. Here, an ER network has a uniform distribution. Second, where Pi,j values are along the x-axis and their frequency along the y-axis, as in Figure 6. Here, an ER network shows a single δ peak -- as far away from a uniform distribution as possible.

---

## [Author Response]

Essential Revisions:Both Reviewers appreciate the scope and importance of the study in both experimental and modelling components. There was also a consensus among Reviewers that it is necessary to1) clarify the separation between functional and structural connectivity;

This is an excellent point, which also echoes points raised by both reviewers in their public reviews; thank you for pushing us to clarify. We have focused on functional connectivity rather than structural connectivity. We had previously placed this in the Discussion, but we have now clarified in multiple places the distinction between functional structural connectivity:

We updated a caveat at a key part of the early discussion:

[Page 26] “It is important to note however that the present study focuses on functional connectivity, rather than structural connectivity, and this separation is discussed in more detail later in our limitations.”

We then added in key clarity in line with Reviewer 1’s public suggestions:

[Page 28] “Nevertheless, it is important to keep in mind how our presented results and interpretations may or may not generalize to structural connectivity graphs of in vitro neuronal networks directly, despite the same rules approximating macroscopic structural connectomes. For example, the type of cost incurred by the existence of a long-range functional connection is not necessarily equivalent to that incurred by a structural connection, as a *direct* physical connection may not exist. Future work is needed to investigate directly structural connectivity at this scale and examine the extent to which there are equivalences between the formation and development of structural components (e.g., synapses and axons) alongside their functional interactions.”

We also have clarified it is *functional* topology in line with Reviewer 2’s public suggestions, adding in the term whenever it was possible:

[Page 20] “Despite alterations in cellular activity and functional network topology (Figure 6— figure supplement 1 and Figure 6—figure supplement 2), homophilic generative attachment rules were the best fitting models across both control, gabazine-treated and washout conditions (Figure 6—figure supplement 4a,b). However, gabazine cultures exhibit a higher energy relative to controls (p=0.0292; Figure 6c), which suggests that homophilic GNMs cannot approximate the functional topology of gabazine-treated cultures to the same extent as for control cultures. This finding supports our hypothesis that perturbing GABAA receptor-mediated inhibition alters functional network characterization, even though homophily remains the best fitting model.”

2) provide the actual optimized model parameters in addition to the rough overview in plots like Figure 3C;

We are happy to provide all the optimized model parameters. We have now added a new Supplementary file 1c that documents precisely the optimized parameters for each group of cultures across time and group. This means that readers can now cross-reference the parameters directly.

[Page 9] “Foreshadowing our findings, we report all optimized model parameter fits and energy values, across all analyzed datasets, in Supplementary file 1c.”

3) clarify differences between "more structured" vs. "more random" networks

Thank you for highlighting the need for further clarity on this. We very much appreciated both Reviewer 1 and Reviewer 2’s comments regarding this point too. We have now moved away from more broad statements of “structured” versus “random” to be precise that we mean is the specificity of wiring in the generative process. In a situation in which the probability distribution becomes flatter (compared to a positively skewed distribution), when a new connection is formed it is less specific precisely where this connection will emerge. This has also been shown in recent animal work by Carozza et al. 2023, which we now cite. This work shows that as wiring parameters are weakened in magnitude, probability distributions flatten, leading to less certainty in the eventual network outcome. In the situation where there are stronger wiring parameters, there is a greater probability skew on a smaller number of future possible connections (i.e., smaller number of high probability score connections) which increases the relative specificity that this is where the connection will fall in the next timestep.

We’ve updated the main text to be clearer to the readers:

[Page 21] “Therefore, we hypothesized that this would lead to a flattening of this lognormal distribution, meaning that the resultant topology became more random (see Methods; Generative probability distributions for detail). Indeed, in Figure 6d we show this to be the case: gabazine-treated networks exhibit a flattened *P_i,j_* distribution relative to both controls (median *P_i,j_* value=0.135 and 0.322 for gabazine & control, respectively; p=1.54x10^-44^, Cohen’s *d*=0.550) and also after gabazine washout (median *P_i,j_* value=0.179; p=5.04x10^-8^, Cohen’s *d*=0.196; see also Figure 6—figure supplement 4d). This finding suggests that gabazine alters the network wiring distribution as to become more variable in its wiring preferences, rather than being specific to a relatively smaller number of candidate neurons that are deemed particularly valuable to wire with.”

We also took this opportunity to address a comment from Reviewer 1’s public review (but was not in the private comments) which was a clarification query relating to how flattening the probability distribution:

[Page 41/42] “Note that the probability score distribution “flattening” (from a relative lognormal distribution toward a uniform distribution) means that relatively more candidate connections come to have a higher probability of being connected in the future. This leads to a decreased specificity of future wiring (i.e., the scope of possible outcomes increases). This flattening effect is equivalent to the network outcomes becoming more random, because a greater number of possible future connections have a non-trivial probability score, and the formation of a connection is less driven by the underlying network topology. Note, a completely singular and uniform *P_i,j_* distribution (i.e., where all connections have the same probability of wiring) generates an entirely random graph (Erdos & Renyi, et al. 1959). In contrast, when a generative network has a heterogeneous distribution in its *P_i,j_* distribution (e.g., with large numbers of small probability scores and small numbers of large probability scores) the network will have a smaller scope for variability (for more detail, see (Carozza, et al. 2023)).”

References:

Erdös, P., & Rényi, A. (1959). On Random Graphs I. Publicationes Mathematicae Debrecen, 6, 290-297.

Carozza, S., Akarca, D. & Astle, D. The adaptive stochasticity hypothesis: Modeling equifinality, multifinality, and adaptation to adversity. Proceedings of the National Academy of Sciences 120, e2307508120 (2023).

4) make an effort to control for different amounts of data across datasets. For example, for a larger dataset, analyses could be carried out on a subset of the data to show how results might depend on the dataset size.

This is a great point and we are happy to include this in the revised manuscript. All the models are fit at the level of individual neuronal cultures, but practically speaking, it was not possible to generate equal numbers of cultures across all the varied experiments. As noted, this means that whilst all the statistics consider the number of cultures, it is possible that our sensitivity to detect differences varies across the datasets. The largest dataset we have is in the dense rodent PC dataset (n=12 cultures over the time points; 100,000 cells per well plating density). As suggested, we can subsample (at n=6 and n=3, to bring in line with the other datasets) to test the model outcomes (model fits and parameters) sensitivity to dataset size. We find that the results are robust to sample size because the effect sizes are particularly large. We have also validated our analysis in Figure 7—figure supplement 3 (see below).

We outline the key results within a new Supplementary file 1aj.

We outline in the Methods how we conducted this experiment and the interpretation:

[Page 41] “Due to different amounts of data across datasets (e.g., see Supplementary file 1c) we conducted a subsampling procedure in which we sampled from our largest dataset (dense rodent PC cultures) to assess to what extent results may depend on the dataset size. To do this, we took the n=12 cultures across developmental time-points (DIV 14 and DIV 28) and sampled n=6 and n=3 (respectively) samples 1000 times. For each of the 1000 sub-samples, we computed statistical testing on the model energy acquired from all assessed generative models and provided 95% confidence intervals on pairwise comparisons and effect sizes. We report these findings in Supplementary file 1j, showing that even when sampling as low as n=3 our results appear to generalize very well, with homophily remaining as the rule generating the lowest energy. Of note, we find a general trend that lowering the empirical sample size increases the p-value, which can be considered by those conducting future studies.”

For completeness, we have also added in a further analysis to show how our results are consistent across alternative spike-sorting/post-processing pipelines and provided a new Figure 7—figure supplement 2:

[Page 35] “To probe whether the reported main results depend on our spike-sorting/postprocessing pipeline, we also prepared the data using an alternative pipeline used a later version of Kilosort (version 2.5), accessed through SpikeInterface^151^, and the quality control toolbox Bombcell152 to screen for units that may be included in the analysis.”

Moreover, we restricted functional connectivity inference to unit pairs with a minimum of co-activity, i.e., at least 50 spikes within a ±50 ms time window of the cross correlogram calculated between the units. We also excluded neuron pairs with potential spike sorting issues, slightly modifying the spike sorting index introduced by a recent study to our data153. As we show in Figure 7—figure supplement 2, results from this analysis were mostly in line with the initial results reported in the main manuscript.

Moreover, we replicated our analysis on another, independent dataset of sparse rodent PC cultures (n=12; plating condition: 50,000 cells per well; recording time point: DIV 14). We added a short paragraph on this dataset to the Methods.

[Page 32] “Rodent primary cortical culture validation dataset

To probe the reproducibility of our results, we obtained a second set of rodent PC cultures. The protocol to prepare this dataset was very similar to the procedures described previously for our main datasets. The wells of a 24-well HD-MEA plate (MaxWell Biosystems) were sterilized in 70% ethanol for 30 minutes and rinsed three times with sterile water. To enhance cell adhesion, the electrode area of the HD-MEAs was treated for one hour at room temperature with 20 µL of 0.05% (v/v) poly(ethyleneimine) (Σ-Aldrich, #181978) in borate buffer (Thermo Fisher Scientific, #28341) at 8.5 pH, and then washed three times with sterile water. Next, 10 μL of laminin (0.02 mg ml^−1^; Σ-Aldrich, #L2020) in Neurobasal medium (Gibco, Thermo Fisher Scientific, #21103049) was pipetted on each array and again left for one hour at room temperature. Rodent PC neurons were prepared and plated as previously described (Ronchi, et al. 2019). The plating density for this dataset was 50,000 cells per well (1000 cells/mm^2^); the recordings included in the replication analysis were performed on DIV 14. The results for this analysis are depicted in Figure 7—figure supplement 3.”

Reference: Ronchi, S. et al. Single-Cell Electrical Stimulation Using CMOS-Based High-Density Microelectrode Arrays. Front. Neurosci. 13, (2019).

The results of this replication/validation analysis are presented in Figure 7—figure supplement 3.

To point this to readers more clearly, in the main text, we have added:

[Page 24] “We replicate our main findings using alternative spike-sorting and post-processing pipelines and validate on an independent dataset in Figure 7—figure supplement 3 and Figure 7—figure supplement 3 respectively.”

Reviewer #1 (Recommendations for the authors):First a list of apparent inconsistencies and minor points of the presentation:Figure 2b: Is reported to be from network recordings, but seems to have only ~28 nodes based on the histograms. That is lower than reported in S1.

We apologize for this ambiguity. The caption was not meant to imply that the presented graph in Figure 2b was recorded. It was only meant to describe what was done in the paper. The image in Figure 2b is a subpart of an empirical graph just to demonstrate graph-theory concepts to the readers. We’ve now made this clearer by updating the caption:

[Page 9] “Each panel on the top row shows an example schematic network, with the node size corresponding to the respective graph statistic (degree, clustering, betweenness centrality, total edge length).”

Figure 2b: Why are values for betweenness > 100? Would not expect that, given the equation in the methods. Looks more like the total number of shortest paths is reported, rather than a fraction.

Thank you for picking up this error on our part. You are correct. We had quoted a normalized version of the same measure. We have now updated equation 7 to remove this normalization to now provide:\begin{document}$b_{i} = \sum_{h,J \in N}\frac{\rho_{\text{hj}}(i)}{\rho_{\text{hj}}}$\end{document}

Figure 2d: The list puts models 3-7 as degree-based, and 8-12 as clustering-based; the boxplot has it inverse.

Thank you for picking up this error. We have now corrected this in the new Figure 2d plot.

Figure 3f: Maybe more of a question: Why does DIV 7 correspond to 50% of "simulated time"? According to Supp Figure S2, DIV 14 has a median density of around 7% and DIV7 has < 1%. Is simulated time not the fraction of edges placed?

This is a great question. Rather than aligning the simulations of the culture on the plot with respect to the number of connections, we aligned the cultures with respect to the *age* of the cultures. As DIV 7 is half the age of DIV 14 we compared them to the half-way point of the simulations of DIV 14. We did this, rather than with respect to the number of edges, so that we could more easily align across which will have the same DIV but have subtly variable numbers of connections (within- and between- time points). We have added this consideration in the Methods section to ensure clarity for readers:

Page [13/14] “Results were scaled to the age of the culture so that observations and simulations could be compared directly, where 50% of the simulated time corresponds to DIV 7 and 100% of simulated time corresponds to DIV 14. Note that simulated time here refers to the percentage completion of the simulation, rather than the number of connections added, so that cultures could be aligned based on time. The quoted r and p value correspond to the correlation between observed and simulated statistics.”

Figure 5: What exactly is "matching" that is depicted in the 6th row and column of the correlation matrices? It is explained in the methods as a measurement related to pairs of nodes, but I would expect a per-node-level measurement.

This is a very good point, and we apologize for not being explicit. It is indeed a measurement related to a pair of nodes. Put simply, the matching value corresponds to the normalized overlap in connections in the neighborhoods of two nodes. But, as you allude to, all measures in the correlation matrix represent node measures (not a node-pair / edge measure). Here, matching is the *average* matching index between its node-pairs, making it a node measure. This is why it can be used in the topological fingerprint calculation. We’ve now made this clear in the caption for readers:

[Page 16] “Each node-wise measure used within the correlation matrix is plotted (left). As matching is an edge-wise measure, the matching calculation presented on row six was derived as a node-wise average.”

Figure 6: Aren't the P_ij values a bit high? The way I understand the generative algorithm after reading this manuscript and Akarca et al., 2021, is at each step a single edge is selected and placed according to P_ij. So the value should add up to 1.0. Even if more than one is selected at a time, with a median value around 0.2, we would reach an edge density of 0.2 in a single step, so that can't be it.

While the P_ij does refer to a probability distribution, it is better described as a probability *score* distribution rather than a strict probability score that sums to 1. This is because it is a calculated probability score distribution that determines the likelihood of it forming a connection in the timestep. We agree that we should make this clear to the reader. We have now added the following:

[Page 8] “*P_i,j_* reflects the probability score of forming a fixed binary connection between neurons *I* and *j*. This is proportional to the parametrized multiplication of costs and values. Two wiring parameters, η and γ, respectively parameterize the costs and value terms, which calibrate the relative influence of the two terms over time on the likelihood of forming a connection in the next step.”

and

[Page 39] *“P_i,j_* reflects the probability score of forming a fixed binary connection at the current time step. Note that, as this is a probability *score*, the sum of the values may not necessarily equal one. This is because the score indicates the relative likelihood of nodes forming a connection over others.”

Figure 7: Caption for panels b and c swapped

Thank you for pointing this out. We have now corrected this.

Line 592: The case of "no principle being implemented" is listed as being indicated by high energy for all models. Yet, random null models in S5b all have very low energy. (As expected, just set both wiring parameters to 0.)

This is a good point and highlights that our phrase was not universally accurate. We have updated this language to be specific that they would have the same energy (irrespective of high or low):

[Page 19] “preventing any clear connectivity principle being implemented above the others (where all models have the same energy).”

Line 1198: The explanation of global efficiency as depicted uses Euclidean distance (d_ij) and is independent of the graph structure.

Thank you for pointing this out. Our writing was confusing because we had defined d_ij above as the Euclidean distance for the purposes of the computational model, but when calculating the graph theory metric ‘efficiency’ we mean shortest path lengths. We have clarified this now by adding the following below the global efficiency equation:

[Page 37] “where here, *d*_I,j_ represents the shortest path length between *i* and *j*. \begin{document}$d_{i,j}^{- 1}$\end{document}, therefore, represents the inverse shortest path length, meaning that a high efficiency corresponds to a low shortest path length.”

Global efficiency is explained in the methods, but not the local efficiency that is used in Supp Figure S6.

Thank you for pointing out the missing equation. We have now added the following full

explanation of local efficiency:

[Page 38] *Local efficiency.* The local efficiency of the network is given by: \begin{document}$E_{\text{loc}} = \frac{1}{n}\sum_{i \in N}^\frac{\sum_{j,h \in N,j \neq i}a_{\text{ij}}a_{\text{ih}}\left\lbrack d_{\text{jh}}\left( N_{i} \right) \right\rbrack^{- 1}}{k_{i}\left( k_{i} - 1 \right)}$\end{document} where E is the local efficiency of node i, and d(N) is the length of the shortest path between *j* and h, that contains only neighbors of *i*.

Supp Figure S11b: I am not sure I understand the plot. the x-axis is labeled "log10", but the tick labels are also logarithmically spaced. Also, is the unit really in ms?

We apologize for the confusion. We have plotted the log (base 10) scaled ISI that were measured in ms. But we understand that because the x-ticks are logarithmically scaled (rather than the power) this would cause confusion. We have instead now replaced the x label to simply state ISI (ms). Now it should be clear that we have plotted a histogram of the ISI (in ms) but transformed onto the log scale, and this is consistent with the caption.

General notes:Overall, a number of powerful and interesting methods and analyses are employed. But their selection seems a bit random. Between Figure S6 and S9 "participation" is swapped for "matching". The "topological fingerprint" is used to evaluate model match for sparse and dense cultures, but not for the GABA block case. In the GABA block case, the internals of the generative model is inspected, but not for dense vs sparse. And more. This makes the whole work seem disconnected.

Thank you for this comment. We appreciate that some aspects of the paper may seem disconnected due to the extent of the analyses/datasets. We hope our changes above and below have addressed/will address these concerns, as we think this has better systematized the study.

I think Figure 5 belongs before Figure 4 as it provides further data supporting the idea that the "matching" algorithm builds networks comparable to the in vitro data. Branching off to the case of denser cultures is separate from that.

On reflection, we agree. We have now swapped these sections directly. Thank you very much for this suggestion.

If I understood correctly that different wiring parameters (η and γ) are optimized for the same model but different cultures of the same type, then it should be attempted to explain the differences. (Not just for the GABA block case.) Do they depend on the size of the model? Age? Density? Average activity level?

This is a good point. Each η and γ are optimized for each individual culture across the 13 different models. So you are right that we could do better in explaining the differences in η and γ as a function of the age, density and other relevant factors present in the networks. We’ve added the following new Figure 3—figure supplement 3 which presents this analysis for the sparse (50,000 cells per well plating density) rodent data over its development. We now show that all non-homophily models are highly dependent on the network properties, but this isn’t the case with homophily models. This means in our sample, for example, that homophily does equally well at approximating observed networks across time, and if the networks are larger/smaller and have higher/lower correlated activity. But these factors make other models worse. We’ve added the following text to point readers to our analysis, which can be found now in the new Figure 3—figure supplement 3:

[Page 11] “This model, in contrast to non-homophily models, uniquely generates low model fits and optimized model parameters independent of the increasing correlated activity and changing topology of the networks over this developmental period (see Figure 3—figure supplement 3 for detail).”

I am curious why the main point of the paper is based on the sparse cultures and the dense cultures are much less prominently featured, although the sparse case has n=6 and the dense case n=12.Overall, I am surprised by which data are shown in the main figures, and which ones are supplementary.

We appreciate your comment about what is shown in the main versus supplementary figures. Initially, we only had the sparse network data for earlier developmental times (DIVs 7-14), and we found it an interesting observation that the energy value landscapes emerged as networks developed (whereas after DIV 14, these values seemed stable). Moreover, due to the length of the material in the manuscript and supplement, it has been a real challenge to curate what should show up in the main versus supplementary reports. We wanted to construct the paper to provide the key message that homophily models best approximate in vitro neuronal networks because we think this will appeal to the broadest audience; while still allowing those from different disciplines to explore specific aspects that they may find most interesting in the supplement, in depth. We hope our responses have facilitated this for you.

Reviewer #2 (Recommendations for the authors):General:I am a little confused about how the statistics of the different generative networks are computed. Sometimes it mentions "top performing simulated networks", sometimes "top 10" or "top 50", others "top n=1", and "top n=99 and average". I do not understand why several results are based on the "top n=1 performing simulated networks". These are probabilistic models for small networks, how is the best simulation representative of the underlying wiring principle? The original methodological paper (Betzel, Neuroimage 2016) appears to report the top 1% of energies (100 networks).

Thank you for bringing up this good point. There is some variability in the literature about what to report for the best performing parameters in these generative models. To improve the clarify, we made the following changes:

1. We have now kept the nomenclature consistent to additionally state “top n=1” rather than just saying “top performing simulated networks”. This occurred three times in captions, so hopefully this now clarifies precisely what we mean by our statement “top performing simulated networks”.

2. The reason we provided numerous numbers of top performing simulations was because we wanted to show consistency across numerous numbers of top performing parameters (e.g., n=1, n=10 and n=50) rather than just show it at a single threshold (e.g., top 1% of energies as in Betzel et al. 2016) which may present a potential bias (due to an accuracy-precision trade-off). An alternative approach would be to re-run the generative landscapes for each culture multiple times and average the top n=1 best simulation across these landscapes. This method was put forward since our submission by Liu, et al. 2023 NeuroImage (https://www.sciencedirect.com/science/article/pii/S1053811923001088). However, this approach requires all cultures to have the same number of nodes (as in human in vivo imaging) but this is not possible with HD-MEAs. We have outlined this in our limitations which we hope clarifies this:

[Page 40/41] Our search was intentionally extensive such that we had greater likelihood of converging on accurate parameter estimates for each culture with the capacity to generate good model fits. In the main paper, we report the top performing n=1 parameter simulation, but in each case, we replicate our findings across a variable number of high performing parameters (n=10 and n=50). This was done to show robustness to possible model stochasticity, which might have biased our results. Of note, recent work has provided a new solution to enable easier convergence on true parameter estimates in generative models (Liu, et al. 2023), but this approach requires the sample to have the same geometric structure which is not the case across our HD-MEA recordings.

Reference: Liu, Y., Seguin, C., Mansour, S., Oldham, S., Betzel, R., Di Biase, M.A., Zalesky, A. (2023). Parameter estimation for connectome generative models: Accuracy, reliability, and a fast parameter fitting method, NeuroImage, 270, 119962, https://doi.org/10.1016/j.neuroimage.2023.119962.

3. Across the whole paper, when presenting model energies acquired from the generative modeling, we had reported the main n=1 top performing simulation but then also provided additional simulation results (in the supplement) across the top n=10 and n=50 simulations to demonstrate robustness. However, we did find a single case where we only provided the top n=1 simulation, but not the others. This was for Figure 3—figure supplement 3. We’ve now extended Figure 3—figure supplement 1 to show, as in all other cases, that the results are consistent:

Similarly, it is often difficult to understand how significance was assessed across many figures and tables. For example for the anovas, and cohens effects on the energies. Sometimes energies are presented with 4 groups, others with 13. It is unclear if the same procedure was always followed.

We are happy to clarify our procedures. We have now updated our Methods to make it clear that we have reported all rule-based statistical comparisons (4 groups) for every analysis, but that we also provide Supplementary Figures showing the model fits by each individual model type too (of which there are 13):

[Page 39] When we report model fits in the main report, we group the model fits based on the broad generative rule group (as shown by the distinct colors in Figure 2d). Supplementary file 1c contains a summary of all optimized parameters and model fits. Supplementary files 1d-h show all statistical comparisons (One-way ANOVA and pairwise comparisons) between generative rule groups, over all time points for each considered analysis.

Line-based:159: It is very difficult to relate the reported plating densities with densities reported in other studies. It'd be much better to report real densities (in cells/ mm2) measured at the time of recording (or after staining). See for example the works by Potter, Segal, Moses, labs, and others.

We thank the reviewer for bringing up this point and agree that the description of the cell densities could have been clearer. In the revised manuscript, we have added such details in the Supplementary file 1a.

We also agree that providing direct estimates of cell counts on the recording day would be more informative, as it is well established that not all neurons initially plated survive to the later developmental time points. Although tracking neurons across development on our chips might have been possible with reporter lines (e.g., that express a fluorescence tag in neurons for livecell imaging), we did not have such lines at hand. Moreover, another complication is that the used HD-MEAs are non-transparent. Nevertheless, to address this important comment, we added a sentence to the Methods

**[**Page 31/32] Notably, cell densities were estimated only on the day of plating and not on the actual recording days. As reported in previous research (Xing, et al. 2021), it is assumed that the actual cell numbers on the HD-MEAs, on the respective recording days, were significantly lower than the initially plated numbers. This reduction may result from incomplete adherence to the HD-MEA, stress during the plating process, the composition of plated cells, and/or naturally occurring cellular processes such as apoptotic cell death.

Reference: Xing, W., de Lima, A. D. & Voigt, T. The Structural E/I Balance Constrains the Early Development of Cortical Network Activity. Front. Cell. Neurosci. **15**, (2021).

171: After the whole-array activity scan, how were the 4x4 blocks selected? I assume one ends up with 64 independent 4x4 blocks, was the distance between them consistent? Was the ranking procedure from 1103 also used? Given that each 4x4 block can identify more than 1 cell, it is unclear why the number of tracked units in the sparse and dense cultures is so similar. Could it be that the electrode selection procedure was biased?

The electrode selection procedure for the network recordings with 4x4 electrode blocks is based on the whole-array pre-scan, the so-called activity scan. The electrode selection for these blocks applied the “Neuronal Units” option in the “Network Assay”, as provided by the MaxLab Live software (MaxWell Biosystems). It combines values, on either the detected spike amplitude or activity of electrodes, and a minimum distance between the centroids of the blocks. This minimum distance between the centroids, however, does not guarantee that all blocks are equally far apart. Because the used HD-MEAs do not allow for full-frame readouts, the selection of electrodes is focused on neurons which exceed a predefined activity/amplitude threshold – hence, there is clearly a selectin bias. The actual number of blocks and the number of connected read-out electrodes, however, also depends on other factors such as the location of the centroid of each block on the chip, and how well the switch-matrix routing procedure was able to connect electrodes for readout.

We clarify this now in the Methods:

[Page 34/35] “To characterize rodent in vitro neuronal networks on HD-MEAs, and to track their functional connectivity, we performed developmental recordings, starting one week after the plating. Using the MaxLab Live software (MaxWell Biosystems), we first used the “Activity Scan Assay” module, which performs a series of high-density recordings, to screen for active electrodes across the entire HD-MEA. During the activity scan the multi-unit activity for each electrode was estimated by applying an online sliding window threshold-crossing spikedetection algorithm. After the activity scan, we used the MaxLab Live “Network Assay” module to select up to 1024 readout-electrodes based on the identified active electrodes. To track networks at single-cell resolution over development, we obtained high-density network recordings, consisting of up to 64 non-overlapping 4 x 4 electrode blocks (electrode-to-electrode pitch of 17.5 μm), and a minimum spacing of at least 87.5 μm between the centroids of the blocks. Depending on the dataset at hand, the block configurations were based on an activity scan performed on DIV 7 (sparse PC networks) or DIV 14 (dense PC networks). The duration of the HD-MEA network recordings was 30 minutes (for each day); an overview on the different datasets is provided in Supplementary file 1a. The sparse/dense rodent PC neuronal networks (main manuscript) and the hCO’s were recorded on MaxTwo 6-well plates (MaxWell Biosystems); the human iPSC-derived neurons (glutamatergic, motor and dopaminergic neurons) were recorded on single-well MaxOne HD-MEAs (MaxWell Biosystems; see also (Ronchi, et al. 2021)). Finally, the validation dataset (Figure 7—figure supplement 3), comprising sparse rodent PC neuronal networks, was acquired on 24-well plates (MaxWell Biosystems).”

Reference: Ronchi, S. et al. Electrophysiological Phenotype Characterization of Human iPSC Derived Neuronal Cell Lines by Means of High-Density Microelectrode Arrays. Advanced Biology **5**, 2000223 (2021).

Importantly, the number of tracked units also depends on other factors, such as how well the spike sorting worked and how many cells met the criteria set during the post-processing and quality-control operations. All these factors contribute to the number of units inferred from the dense/sparse primary networks – and impact the size of the network. However, since our results generalize across densities, we do not regard this as a major limitation. Our results also replicate across different cell lines, which were not based on the 4x4 block recordings, but an electrode selection scheme that used the top 1024 most active electrodes (human neurons). Nevertheless, we agree with Reviewer 2 that studies should investigate how the (sub-)sampling of the underlying network affects the inferred connectivity/topology and the resulting generative model fits. We added a brief paragraph on this aspect to the Discussion/limitations section:

[Page 28/29] “Another important consideration is the impact of subsampling, and specifically, how the selection of HD-MEA recording configurations and neuronal units (e.g., including units only above a specific activity threshold) impacts the reported organizational properties in functional connectivity. Previous studies established that subsampling can significantly affect the inference of network dynamics and connectivity (Das, et al. 2020, Pillow, et al. 2007), and this should be considered when interpreting our results. Future studies could further probe how different parameter choices for selecting HD-MEA recording configurations and the exclusion of low activity units alters neuronal topology. Similarly, it would be important to test if the reported results also hold for more advanced network inference algorithms (Donner, et al. 2024).”

References: Das A, Fiete IR. Systematic errors in connectivity inferred from activity in strongly recurrent networks. Nature Neuroscience. 2020;23(10):1286–1296. pmid:32895567

Pillow JW, Latham P. Neural characterization in partially observed populations of spiking neurons. Advances in Neural Information Processing Systems. 2007;20.

Donner, C. *et al.* Ensemble learning and ground-truth validation of synaptic connectivity inferred from spike trains. *PLOS Computational Biology* 20, e1011964 (2024).

190: Regarding the jittering procedure, it is unclear whether it involved a random shift of either + or – 10 ms? or any value within that range (uniform distribution?)

We thank the reviewer for this comment and improved the description of the jittering procedure in the revised manuscript:

[Page 36] “For a given neuronal unit’s spike train, spike times were randomly jittered by sampling from a normal distribution with a standard deviation of 10 ms, generating a surrogate spike train. The code for jittering spike trains was adopted from the Neural Complexity and Criticality Toolbox^156^. The jittering and STTC inference procedure was repeated for each neuronal unit for 1000 permutations.”

310: Based on all the energy/exponent maps (e.g., Figure 2d right), the relevant range of exponents is always between -1 and 1 (or -3,3 at best). It is very difficult to read the value of the exponents based on the actual maps (e.g., Figure 3c). Maybe the authors could report the actual exponents somewhere (as they did in Sup Figure 14c for example). These values might help the interpretability of the results (for example, whether the exponents are positive or negative could be quite relevant).

This is a great suggestion, thank you. We have now added a new Supplementary file 1c that documents precisely the optimized parameters for each group of cultures across time and group. This means that readers can now cross-reference the parameters directly. For example, you can now clearly see whether the parameters are positive or negative.

[Page 8] “Foreshadowing our findings, we report all optimized model parameter fits and energy values, across all analyzed datasets, in Supplementary file 1c.”

357: What "large topological changes" do the authors refer to? Figure S2 only reports an overall increase in the number of connections. Also, is the network density measured as "total degree / (1/2*N*(N-1))" ? Or is there an extra factor of 2 somewhere? I'm trying to relate the values in S2a S2c, and supp Table 1.

We have now made this statement more specific to what we meant by “large topological changes” in terms of an increasing number of connections and network density:

[Page 11] “As previously shown (Figure 1), PC rodent networks underwent significant developmental changes during this time period. These changes are reflected by large changes in their functional networks, with an increasing number of connections and network density (Figure 1—figure supplement 2).”

On your second point, indeed the density is calculated as the percentage of present connections relative to all possible connections, which is exactly as you describe as a percentage: 100*total degree/(0.5*n*(n-1)). However, this is a graph measure of the network rather than the plating density quoted in Supplementary file 1a which we think may have caused the confusion. We’ve updated the caption to Supplementary file 1a to clarify this point:

[Page 28, Supplement] “The plating density here reflects the number of cells plated, rather than the graph density of the connectivity of the functional networks that are subsequently inferred following developmental time.”

374: Regarding "homophily performs best when approximating empirical networks, but not randomized networks", how did the authors measure that? The full random case in S5b has the lowest energy of all, and lower energy would mean a better fit if I'm not mistaken. Or were the authors only talking about the "randmio" procedure? If so, the authors might want to plot S5b in the same range as the figure it has to be compared with (0 to 0.6).

Thank you for this point. The crucial clarification we needed to give is that we mean the *relative* performance of homophily to approximate the topology. For example, in S5b left we see that random networks can be simulated by all of the generative models simply because both wiring parameters can be tuned to zero – which is equivalent to random wiring. To make this clear, we have updated the S5b panel caption to make this clear:

[Page 10/11, Supplement] “In the case of completely random networks (left), there is no difference in relative performance across wiring rules, because the optimal parameter fit is achieved when both wiring parameters are set to zero, as this generates a totally random topology regardless of the rule.”

We’ve also updated the main text too:

[Page 11] “We find that, regardless of the model specification or connectivity measure tested here, homophily performs best when approximating empirical networks relative to all other models, but not randomized networks (Figure 3—figure supplement 4).”

We’ve kept the y-axis from 0-1 in this figure just to allow all null models to be compared on the same scale. We hope that our above changes have clarified your point.

375: I don't understand how the staining panel relates to the rest of this figure.

This staining panel is just to provide the readers with a visualization of some of the data that was recorded. When sharing our work with others, several people asked to see something like this to give them an intuition of how the cultures look. Unfortunately, outside of our description, we could not find an appropriate place to explicitly state this.

376: F3C, to which of the 13 generative models does this plot correspond to? If I'm not mistaken, the exponents are calculated independently for each of the 13 models and 4 time points.

The current caption states: “All energy landscapes of the matching generative network model for DIV 14”. We’ve now changed it to the following to make this clearer:

[Page 13] “All energy landscapes of DIV 14 networks, plotted on top of each other, under the matching generative network model (a homophily model).”

415: Unsure what the R2, r and p measures mean. Is this just pearson on F3e,f right panels, and its p-value?

Yes, the reviewer is correct. We apologize that this was not clear. We have updated the caption for F3e,f:

[Page 14] “The quoted r and p value correspond to the Pearson’s correlation between observed and simulated statistics shown.”

442: Same as the comment on L159, the authors should report the density at the time of recording. It is also worth noting that there are many more processes affecting the final density than just apoptosis.

We agree with Reviewer 2 and have added a paragraph to the Methods and a Supplementary file (see reply to L159). We also refer the reader to a study that probed the development of primary neuron numbers on transparent MEAs during a similar developmental period and at a similar plating density (Xing et al., 2021: https://doi.org/10.3389/fncel.2021.687306). Moreover, we now mention some of the other factors that very likely contribute to a decrease in neuron numbers at later time points (e.g., incomplete adherence to the HD-MEA, and stress during plaiting). See reply to L159 and reply on page 31/32 (of the manuscript).

444: I'd like to point out a few important references on this matter. Ivenshitz and Segal, J Neurophysiol 2010. Cohen et al., Brain Research 2008. They looked in detail at how different densities affect activity and network connectivity in cultures.

Thank you for pointing out these references. We have now included these references in the relevant section for readers to refer to.

455: Related to L171. I have difficulties understanding the relationship between plating densities, the number of recorded cells, the distance between cells, and inferred connectivity. From S3, the distribution of Euclidean distances for the two plated densities looks very similar. And the number of tracked units in S1 is also very similar. However, S7 reports functional connectivity with smaller edge lengths and higher clustering. This would be consistent if the higher plating density just resulted in more "cells/mm2" and the activity being invariant. But that doesn't seem the case. The authors should clarify that.

This is a very good point that we did not clarify appropriately in the paper. S3 reported the interneuronal Euclidean distances between all neurons that exist: i.e., the total distances existing between neurons irrespective of the inferred network. This is not the same as the empirical edge lengths of the network which, in (now Figure 5—figure supplement 1), reports the empirical edge lengths of the connections. This means that in the higher plating density the distribution of space between neurons doesn’t change, but the inferred functional connectivity length does. We have now updated this in the caption of both Figure 2—figure supplement 1 and Figure 5—figure supplement 1 to clarify this point.

In Figure 2—figure supplement 1, we’ve added:

[Page 6, Supplement] “For all distributions, we plot the total set of distances between neurons, irrespective of the inferred functional network.”

In Figure 5—figure supplement 1, we’ve added:

[Page 15, Supplement] “Note here that the edge length refers to the average length of existing functional connections in the network. This measure contrasts with Figure 2—figure supplement 1 which shows the inter-neuronal Euclidean distances between all neurons irrespective of the inferred functional network.”

509: Why did the authors choose a dissimilarity measure based on the Euclidean norm? Distances between correlation matrices are non-euclidean (see for example Venkatesh et al., Neuroimage 2020).

Thank you for this reference, as we were not aware of this paper. Here, the Euclidean norm is taken simply as a composite dissimilarity measure between the two matrices derived from the correlation structure of the network properties, rather than functional correlation matrices in as in Venkatesh et al. 2020 – which as you say is non-Euclidean in geometry. In this work, because we are measuring competing model performance, we believe the approach remains valid, but we acknowledge that alternative types of model fitting procedures could be used and should be the subject of future study. We have now added this caveat in the Methods section, and we thank you for pointing us to this:

[Page 39/40] “Moreover, there are alternative procedures to the one provided that could be taken to estimate the dissimilarity between the topological fingerprint matrices, that could provide greater capacity to distinguish generative model performances. For example, the Pearson’s correlation (as in representational similarity analysis; Kriegeskorte, et al. 2008) or the nonEuclidean geodesic distance (which better captures the non-Euclidean geometry of functional connectivity in functional magnetic resonance imaging; Venkatesh, et al. 2020) could be used. Future work should examine the advantages and disadvantages of these approaches for the case of generative network models.”

References: Kriegeskorte, N., Mur, M. & Bandettini, P. A. Representational similarity analysis – connecting the branches of systems neuroscience. Front. Syst. Neurosci. 2, (2008).

Venkatesh, M., Jaja, J. & Pessoa, L. Comparing functional connectivity matrices: A geometry aware approach applied to participant identification. NeuroImage 207, 116398 (2020).

625: If I understood this measure correctly, the STP is asymmetric STP(i->j) != STP(j->i). If that's the case, did the putative inhibitory connections cluster around specific input cells? i.e., following Dale's rule? That would be an interesting check.

We agree with Reviewer 2 that testing Dale’s rule would be an interesting check. For the present data, however, we focused only on strong inhibitory connections, i.e. connections that could be clearly picked up in pair-wise cross-correlation histograms. As outlined in previous work (e.g., English et al., 2017: https://doi.org/10.1016/j.neuron.2017.09.033), weak connections are likely missed with the STP approach. The number of putative strong inhibitory connections (in our data) was therefore low, which complicates more detailed analysis on clustering on specific input cells. We therefore leave this interesting idea for more in-depth analysis on inhibitory connections to future research. We clarified and rephrased the corresponding section in the Results, to point to this limitation of STP analyses:

[Page 20] “To further examine the extent to which washing out gabazine returned endogenous inhibitory activity as GABAA receptors are released from blockade, we also computed the spike transmission probability (STP) among neuronal units (Figure 6—figure supplement 3). This cross-correlogram based metric has been used to identify spike transmission, respectively spike suppression, and to infer putative excitatory/inhibitory functional connectivity from on-going extracellular spiking activity^105^. Indeed, we find that washout of gabazine led to a significant increase in the number of putative inhibitory connections (Wilcoxon signed rank: p=0.012; Figure 6b). It should be noted, however, that weaker inhibitory connections are likely missed with the applied STP approach (English, et al. 2017).”

Reference: English, D. F. et al. Pyramidal Cell-Interneuron Circuit Architecture and Dynamics in Hippocampal Networks. *Neuron* 96, 505-520 7 (2017).

904: How was this replication of the results with TE-based methods quantified? S5b only shows a similar trend as those in Figure 3, but no quantification. Was this measured on the sparse networks only? And were the properties of the TE-based network and the STTC ones similar regarding their topological properties and fingerprint?

We previously had not added statistics to the TE-based methods, because we aimed to show that the trend was similar in this TE comparison. However, we have now conducted statistical comparisons alongside a new analysis that we think will provide clarity to readers. Beyond showing the homophily performs best in TE-networks, we have added a new finding to the Supplementary Figure that shows a large correlation in the energy acquired between TE and STTC measures of connectivity (r = 0.849, r = 0.867 and r = 0.901 respectively for symmetrical, out and in transfer entropy respectively). This suggests that irrespective of the measure used to construct the networks (e.g., STTC or TE) there is a clear alignment in the generative outcomes. We’ve added this point to the main report:

[Page 11/12] “We find that, regardless of the model specification or connectivity measure tested here, homophily performs best when approximating the sparse empirical networks relative to all other models, but not randomized networks (Figure 3—figure supplement 4). For instance, in symmetrized TE networks (i.e., keeping only reciprocal connections), homophily models also perform best (p<1.95x10^-6^ for all pairwise comparisons and Cohen’s *d*>1.12 reflecting a very large effect size). Furthermore, there is a clear alignment in the generative properties of the networks irrespective of if they were constructed using the STTC or TE, demonstrated by a very large positive correlation in their subsequent model energies (r = 0.849, p = 8.97x10^-22^, r = 0.867, p = 9.55x10^-25^ and r = 0.901, p = 2.91x10^-29^ when considering TE symmetrical, in or out (directed) connections respectively compared to the STTC; see Figure 3—figure supplement 4d).”

You can see the new panel in the figure.

Note that further to this, as described above, we have now reported all model parameters and acquired energy values for each dataset to provide additional quantification.

1119: How many cultures were treated with gabazine? Here it mentions "Three". But F6 and the text mention n=9 (6 of them with washout).

Sorry for the confusion: n=9 were treated with gabazine, but n=6 of these were then “washed out”. We have now clarified this more clearly in the main text

[Page 19] “To address this, we cultured sparse rodent PC neurons under chronic application of media without gabazine (n=6, control) or with gabazine (n=9 gabazine-treated), a selective GABAA receptor antagonist (see Methods; Pharmacological experiments). Following chronic application of gabazine for two weeks, we washed out gabazine at DIV 14 and performed a final recording at DIV 15 in a subset of six cultures of the gabazine-treated dataset (n=6, termed washout) to determine the extent to which the cultures recovered.”

1172: A significance value of p < 0.01 was chosen for the connectivity. If I understood it correctly, that means, that on average at this threshold, there's a 0.01 probability the observed SSTC value comes from the surrogate distribution. That would result in a 1% network density just from the false positive rate. That's in line with the values reported in S2a at 7 DIV where the network is probably not yet formed. But does that mean that at 14 DIV the expected number of false positives in the inferred connectivity is ~ 20%? Are the inferred networks robust to changes in the binarization threshold?

Thank you for this point. We have now added a new Figure 3—figure supplement 2 that shows our findings are indeed robust to changes in the binarization threshold:

[Page 11] “Figure 3—figure supplement 1 and Figure 3—figure supplement 2 shows comparisons across individualized models and across significance thresholds.”

Figures:Figure 2d. Energy. The Clustering and Degree labels might have been swapped.

Thank you for spotting this. We have now updated this error.

Figure 3c. Is this for a representative network? Or averaged across the networks?

This plot shows the energy achieved from all DIV 14 rodent PC networks.

Figure 6d. How is the shift in the probability distribution related to the wiring becoming more random across all timepoints? Not sure what the authors mean by this.

We apologize for not being clearer. We have now added extra clarity to this, citing recent work for which readers can explore further. Put simply, when generative networks are made to have stronger constraints (i.e., higher magnitude parameters), the space of possible outcomes grows smaller. This is because the probability score distribution (P_ij) for subsequent wiring becomes more skewed (i.e, has lower entropy). This means that there is a *relative* higher likelihood of specific connections becoming wired together as there are small numbers of possible future connections with high probability scores (relative to the larger number of small probability scores). Conversely, networks with weaker wiring constraints (i.e., small magnitude scalar parameters) generate flatter probability distributions (i.e., has higher entropy). This means that when the next connection is forming, it is less certain precisely which connection will form. This is because there is now less of a disparity in the distribution of possible future connections compared to the prior example. This has been shown, for instance, from a recent Carozza et al. 2023 paper, e.g., see the following figure which shows how – as you increase wiring parameters – the dissimilarity decreases:

We have added the following clarifications in the text regarding P_ij flattening:

[Page 41] “Note that the probability score distribution “flattening” (from a relative lognormal distribution toward a uniform distribution) means that relatively more candidate connections come to have a higher probability of being connected in the future. This leads to a decreased specificity of future wiring (i.e., the scope of possible outcomes increases). This flattening effect is equivalent to the network outcomes becoming more random, because a greater number of possible future connections have a non-trivial probability score, and the formation of a connection is less driven by the underlying network topology. Note, a completely singular and uniform *P_i,j_* distribution (e.g., where all connections have the same likelihood of wiring) generates an entirely random graph (e.g., Erdös & Rényi, 1959). In contrast, when a generative network has a heterogeneous distribution in its *P_i,j_* distribution (e.g., with large numbers of small probability scores and small numbers of large probability scores) the network will have a smaller scope for variability (for more detail, see Carozza, et al. 2023).”

References:

Erdös, P., & Rényi, A. (1959). On Random Graphs I. Publicationes Mathematicae Debrecen, 6, 290-297.

Carozza, S., Akarca, D. & Astle, D. The adaptive stochasticity hypothesis: modeling equifinality, multifinality and adaptation to adversity. 2023. *PNAS.*

Figure 6d. There's a mention of SF13e,f, which doesn't exist. Could it be S14c,d?

Thank you for spotting this error. The Supplementary Figure labels were incorrect in that figure caption. They are all correct now. They were indeed 14, not 13.

Figure 7c. Why does the tsne plot only include the dense DIV 28 primary cultures? The other datapoints should also be included, e.g., sparse DIV 14 and gabazine experiments.

We agree and have updated Figure 7c tSNE plot to also include the sparse DIV 14 and gabazine experiments:

Supplementary:Figure 3. left/right panels should be top/bottom panels.

Thank you for spotting this. This is now corrected.

Figure 10. This figure mentions the dense cultures at DIV 14. Why do they report n=6 when dense cultures are n=12? Or is this figure for sparse cultures instead? The caption also mentions "Distributions are plotted for each […]." There are no distributions in this figure.

Thank you for spotting this mistake. It is indeed dense cultures, and thus n=12. We have removed this incorrect last sentence regarding distributions. It now reads:

[Page 14, Supplement] “Using dense PC rodent networks (100,000 cells per well), we show n=156 data points (n=12 cultures x n=13 generative model simulations) corresponding to the top n=1 performing simulation’s energy and its topological fingerprint dissimilarity performances at DIV 14.”

Figure 11. It would be beneficial to also add a representative raster (panel a) for the washout case. Results in c report a very high firing rate but no bursting rate for the washout case. What does that look like in the raster?

Yes, we agree with you. We have now added in a third horizontal panel to show this. The washout temporal raster plot appears to more closely resemble the control recordings, though some of the population bursting that was seen in the gabazine condition can still be seen following washout. This is now shown in the new Figure 6—figure supplement 1, (previously S11):

Figure 12b. The axis should read ISI (it's already in the log scale). Caption mentions a tri-modal distribution for gabazine. Very hard to see when the washout distribution is on top.

Thank you for pointing this out. We have now corrected this (see above).

Figure 12c. Why is the network size so different between controls, gabazine, and washouts? This is never explained, yet several other measures might depend on this value, e.g., total degree.

This is a good question. It is possible that neuronal survival rate decreases with chronic gabazine treatment leading to fewer neurons surviving to the age at which we recorded. To these ends the network size indicates the number of neurons that met the various inclusion criteria, as defined during the postprocessing, that are sufficiently active and that formed connections. GABAA receptor block ultimately leads to fewer neurons satisfying these conditions. A reduction in inhibition might be expected to increase network activity or potentially lead to excitotoxicity in some neurons, though this likely depends on the complex circuitry. We agree that it is unclear how the measures may depend on this value. However, importantly, we have now shown in a new Figure 3—figure supplement 3 that model energy is low, independent of these factors. Nevertheless, we agree that this is a possible confound and have now mentioned this finding in the Results:

[Page 20] “Gabazine-treated cultures showed a reduction in the number of units forming significant functional connections (lower network size, p < 0.001 and total degree, p=0.012). Note that following washout, network size can change as only spiking neurons are counted as nodes for network size calculation. Network size did not increase significantly (p=0.0625) following washout, though there was a small increase in network size in five of the six washed out cultures (by 18.4±6.2 %).”

We have also clarified these quantifications in the methods:

[Page 36/37] “Network size and total degree. The network size here refers to the number of nodes in the network. This excludes nodes that did not form any connections (neuronal units with <0.01 spikes/s already excluded as above). Network size is sometimes referred to as network order in graph theory literature to avoid confusion with number of edges, however, we refer to the total number of edges in the network as total degree.”

And acknowledged this in the discussion:

[Page 29] “It also remains unclear to what extent the smaller size of gabazine-treated networks, compared to controls, affects other network statistics. However, given the homogeneity of activity across neurons in gabazine-treated networks, smaller network size, rather than flattened probability distributions, is unlikely to explain differences in best-fitting model parameters.”

Figure 15. Why were only the n=12 DIV 14 dense cultures included here?

This was the n = 12 DIV 28 dense rodent PC cultures because these were the most mature rodent recordings (the other human data is also DIV 28). We have now stated this in the caption of Figure 7—figure supplement 1 (previously S15):

[Page 23, Supplement]. “28-day-old dense rodent PC networks (100,000 cells per well) are included here for comparison, as this is the latest time point recorded for the rodent PC cultures, and ostensibly the peak maturity in terms of network activity.”

[Editors’ note: what follows is the authors’ response to the second round of review.]

The manuscript has been improved but there are some remaining issues that need to be addressed, as outlined below:Reviewer #2 (Recommendations for the authors):The authors have addressed the points I listed in my original recommendations for the authors. I thank them for the effort.With respect to my point about "random" vs. "specific" wiring in the context of the GABA block I now have an additional concern:I now understand that the Pi,j values are not direct probabilities, but just indicate relative preferences for edge placement. In that case, I am not sure a direct comparison of distributions in Figure 6d, and of their means is very meaningful. If all values for Pi,j of a given model were multiplied by a constant >1 then it would not affect the model wiring, as the relative preferences remained unchanged. But in a plot such as 6d it would stretch the distribution out and increase its mean.Besides, after reading the authors' reply where they elaborate on 6d, I now wonder if there is some confusion about the following two potential ways to plot Pi,j: First, where all potential edges are along the x-axis and Pi,j along the y-axis. Here, an ER network has a uniform distribution. Second, where Pi,j values are along the x-axis and their frequency along the y-axis, as in Figure 6. Here, an ER network shows a single δ peak -- as far away from a uniform distribution as possible.

You raise an excellent point about the interpretation of these distributions. You are correct that the relative preference would indeed remain unchanged if multiplied by a constant. Our point does not relate to the absolute distribution values, but rather to the shape differences between the distributions across experimental groups. These shape differences remain meaningful because all groups undergo identical generative procedures, so any scaling effects would be consistent across conditions. That is, the observed differences in distribution means reflect genuine differences in the underlying generative processes between conditions.

On your second point, you are completely correct there are several ways to plot P_ij,_ and this changes the distribution referred to. Much of this is motivated by the work outlined by Carozza, et al. 2023 – which in a detailed way outlines the randomness of the resulting networks from the generative process. We previously updated our text to reflect the editorial question on this, but we have now added some additional text too reflecting your comment here and also outlining the key distinction between the ways of rendering the distributions to avoid any confusion:

[Page 41/42] “The probability score distributions, P_i,j_, shown in this work are plotted with the score value on the x-axis and the frequency on the y-axis. It is of note that there are alternative ways that one could represent these distributions, such as the edge index on the x-axis and P_i,j_ on the y-axis, which would provide different appearing distributions which should be carefully considered (for more detail, see Carozza, et al. 2023 which provides a detailed analysis of the magnitude of randomness of generated networks under different parameter conditions).”

We also added some additional clarity in the manuscript:

[Page 21] “Therefore, we hypothesized that this would lead to a flattening of this lognormal distribution, meaning that the resultant topology became more random (see Methods; Generative probability distributions for detail). Indeed, in Figure 6d we show this to be the case: gabazine-treated networks exhibit a flattened P_i,j_ distribution relative to both controls (median P_i,j_ value=0.135 and 0.322 for gabazine & control, respectively; p=1.54x10^44^, Cohen’s d=0.550) and also after gabazine washout (median P_i,j_ value=0.179; p=5.04x10^-8^, Cohen’s d=0.196; see also Figure 6—figure supplement 4d). This finding suggests that gabazine alters the network wiring distribution as to become more variable in its wiring preferences, rather than being specific to a relatively smaller number of candidate neurons that are deemed particularly valuable to wire with.”

We hope these revisions and clarifications address your concerns. We sincerely appreciate your thorough and constructive review.